# Promotion of growth factor signaling as a critical function of β-catenin during HCC progression

Eunsun Kim [1,2], Amanda Lisby[3], Connie Ma[1], Nathanael Lo[1], Ursula Ehmer[4], Katharina E. Hayer [5], Emma E. Furth[3] & Patrick Viatour[1,2]

Hepatocellular carcinoma (HCC) is the second leading cause of cancer-related deaths worldwide. β-catenin is widely thought to be a major oncogene in HCC based on the frequency of mutations associated with aberrant Wnt signaling in HCC patients. Challenging this model, our data reveal that β-catenin nuclear accumulation is restricted to the late stage of the disease. Until then, β-catenin is primarily located at the plasma membrane in complex with multiple cadherin family members where it drives tumor cell survival by enhancing the signaling of growth factor receptors such as EGFR. Therefore, our study reveals the evolving nature of β-catenin in HCC to establish it as a compound tumor promoter during the progression of the disease.

[1] Children's Hospital of Philadelphia, Center for Childhood Cancer Research, Philadelphia, PA 19104, USA. [2] Department of Pathology and Laboratory Medicine, Perelman School of Medicine, University of Pennsylvania, Philadelphia, PAa 19104, USA. [3] Department of Pathology and Laboratory Medicine, Hospital of the University of Pennsylvania, Philadelphia, PA 19104, USA. [4] Klinikum rechts der Isar, Technische Universität München, 81675 Munich, Germany. [5] Department of Biomedical and Health Informatics, Children's Hospital of Philadelphia, Philadelphia, PA 19104, USA. Correspondence and requests for materials should be addressed to P.V. (email: pviatour@pennmedicine.upenn.edu)

Hepatocellular carcinoma (HCC) is the second cause of cancer-related deaths worldwide[1,2]. Most HCC patients are not eligible for curative surgery due to the advanced stage of the disease at the time of diagnosis. Therapeutic options are limited in their availability and efficacy, with multi-kinase inhibitor Sorafenib showing survival benefit of only 3 months[3]. Therefore, there is a critical need for new HCC treatments. Large sequencing efforts have shown that most recurring driver mutations in HCC (p53, telomerase, β-catenin, etc.) are shared with other cancer types[4]. However, a recent global cancer gene expression analysis also reveals that HCC transcriptome is strikingly different from other cancers[5]. From this discovery emerges the concept that common driver mutations may have unique functional consequences in the context of HCC. However, the mechanisms of action of the principal drivers of HCC initiation and progression remain elusive.

β-Catenin is a protein that serves two distinct roles depending upon its subcellular location. First, β-catenin in the nucleus is a critical effector of the Wnt signaling pathway[6]. In the absence of a Wnt ligand, β-catenin is rapidly phosphorylated by the destruction complex (comprising glycogen synthase kinase 3β (GSK3β), casein kinase Iα (CKIα), Axin, and adenomatosis polyposis coli (APC)) on four key serine/threonine residues, which primes β-

catenin for degradation[7]. Wnt ligand binding destabilizes the destruction complex, thus promoting β-catenin stability and its translocation to the nucleus where it serves as a co-factor for TCF/LEF (T cell factor/lymphoid enhancer-binding factor) transcription factors. Second, β-catenin at the membrane is a component of the adherens junction (AJ) complex in association with cadherins. Within the AJ complex, β-catenin acts as a bridge between α-catenin and cadherins to maintain proper cytoskeleton structure, cell–cell interaction, and cell signaling[8]. Earlier works suggest that the AJ complex-associated β-catenin represents a reserve pool of β-catenin that can potentially feed into Wnt signaling activity, thereby allowing E-cadherin to modulate Wnt signaling by physically sequestering β-catenin at the membrane[9–11]. However, the mechanism of initial repartition and subsequent transition between these two pools remains poorly understood.

Mutations targeting components of the Wnt pathway are very frequent in colorectal cancer and hepatoblastoma, and aberrant Wnt signaling activation has been established as a dominant oncogenic driver in these tumors[12–14]. Similar mutations targeting Wnt pathway components are also frequent in HCC (15–33% of HCC patients carry activating mutations in *ctnnb1* (coding for β-catenin)[15], while 17% have inactivating mutations in *axin* or

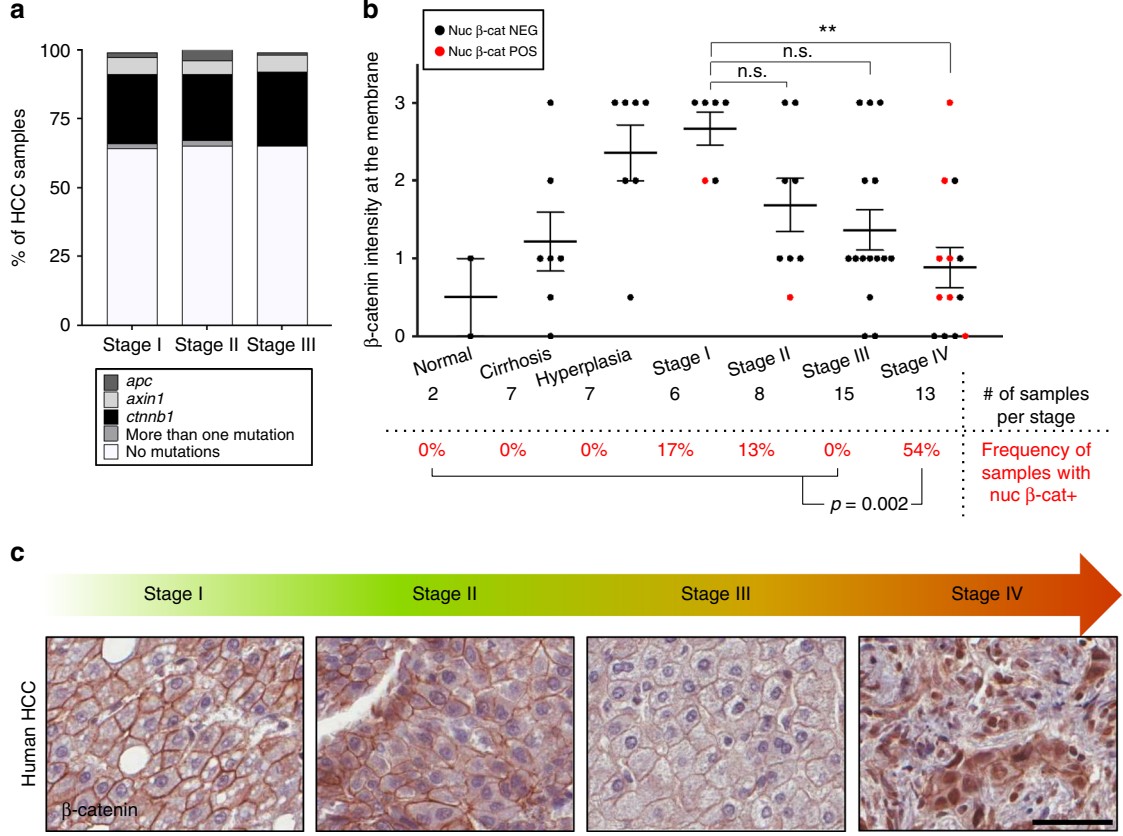

**Fig. 1** β-Catenin is restricted to the membrane until late-stage hepatocellular carcinoma (HCC). **a** Human HCC patient data from The Cancer Genome Atlas (TCGA) database were stratified by HCC stage (I–III) and respective % of samples with one or more mutations in *apc*, *axin1*, and *ctnnb1* (or none of the above) were graphed. **b** β-Catenin localization during HCC progression. A commercial array encompassing the different steps of the disease was stained for β-catenin. A score was attributed to each sample for the intensity of β-catenin expression at the membrane (*y*-axis), with corresponding *p* value between the groups of interest displayed on the top of the graph. Black circles represent samples with an absence of nuclear β-catenin. Red circles represent samples with nuclear β-catenin. The number of samples per stage is provided in the table below the graph. The percentage of patients who display nuclear β-catenin for each stage is displayed in red. The *p* value (=0.002; Tukey's multiple comparisons test) at the bottom of the graph accounts for the difference in the frequency of nuclear β-catenin-positive patients in stage IV vs. earlier stages (stage III and earlier). **c** Immunohistochemistry (IHC) of human HCC tissue array for β-catenin. One representative image is displayed per stage. Scale bar = 50 μm. Data are represented as mean ± SEM. **p < 0.01. n.s. not significant. See also Supplementary Fig. 1

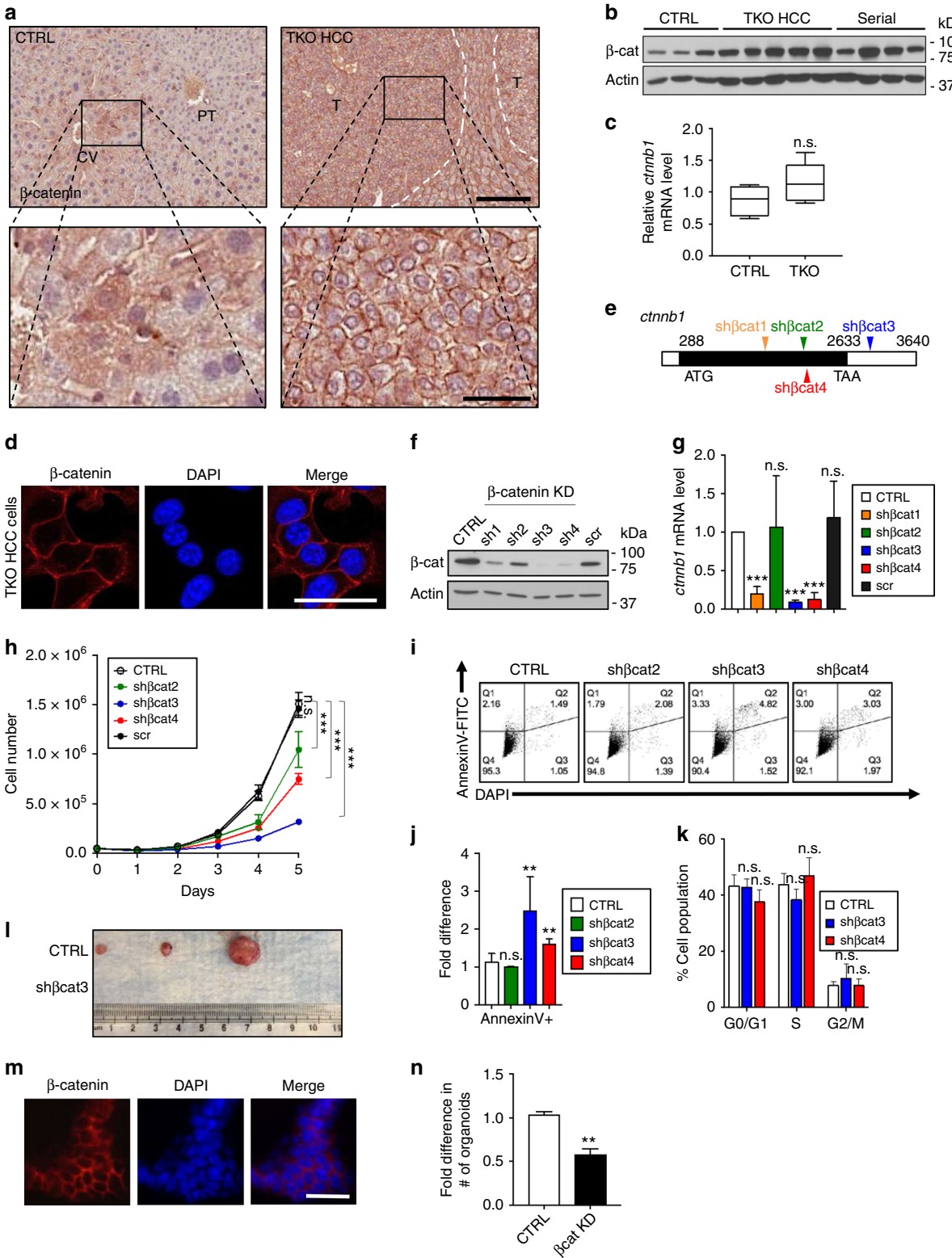

apc[4,16]) and have led to the conclusion that these mutations also act as dominant event to trigger oncogenic Wnt/β-catenin signaling in HCC. However, transgenic expression of degradation-resistant β-catenin in the liver is not sufficient to initiate HCC[17], and it is therefore unclear whether β-catenin activation is a driving mutation or a cooperating event that supports HCC progression initiated by other oncogenic events[18,19]. In addition, loss of β-catenin in chemically induced models of HCC increases tumor progression[20,21], suggesting a paradoxical effect for β-catenin inactivation in HCC. Finally, although the kinetics of Wnt pathway-activating mutations during the course of the disease are poorly known, aberrant Wnt signaling activity is restricted to the advanced stages of the disease[22]. Collectively, these data suggest that the role of β-catenin in HCC is more complex than currently envisioned based on the mutational status of the Wnt pathway in HCC patients.

**Fig. 2** Membrane-localized β-catenin promotes hepatocellular carcinoma (HCC) growth. **a** Immunohistochemistry (IHC) of control (CTRL) mouse liver and triple knockout (TKO) HCC for β-catenin. Scale bar = 100 μm (top), 25 μm (bottom). **b** β-Catenin expression in CTRL livers (n = 3), primary TKO HCC (n = 5), and serially transplanted TKO HCC tumors (Serial) (n = 4), as determined by immunoblot. **c** Quantitative reverse transcription PCR (RT-qPCR) analysis for *ctnnb1* messenger RNA (mRNA) levels in CTRL (n = 4) livers and TKO HCC (n = 9). **d** Immunofluorescence (IF) of TKO HCC cells for β-catenin (red) and 4′,6-diamidino-2-phenylindole (DAPI) (blue). Scale bar = 25 μm. **e** Graphic representation of shβcat1–4 indicating the targeted region within *ctnnb1* mRNA. **f**, **g** Knockdown (KD) efficiency of the shβcat1–4 compared to the empty vector (CTRL) or vector expressing a scrambled hairpin (scr) was determined by **f** immunoblot and **g** RT-qPCR (n > 3). **h** Growth curve of TKO HCC cells upon β-catenin KD with shβcat2–4 (n > 4). **i** Representative fluorescence-activated cell sorting (FACS) plot for AnnexinV-FITC expression on TKO HCC cells upon β-catenin KD with shβcat2–4 (n = 3). **j** AnnexinV-FITC analysis of TKO HCC cells in upon β-catenin KD with shβcat2–4 (n = 3). **k** Bromodeoxyuridine (BrdU) incorporation analysis of TKO HCC cells upon β-catenin KD with shβcat3 and 4 (n = 3). **l** Subcutaneous tumors from nonobese diabetic/severe combined immunodeficiency (nod/scid) mice injected with TKO HCC cells expressing CTRL or shβcat3. **m** IF of TKO HCC organoids for β-catenin (red) and DAPI (blue). Scale bar = 25 μm. **n** Fold difference in the number of TKO HCC organoids upon β-catenin KD with shβcat3. Data are represented as mean ± SEM. **p < 0.01 and ***p < 0.001. n.s. not significant. See also Supplementary Fig. 2

β-Catenin accumulates at the plasma membrane of HCC cells in the absence of significant Wnt target gene activation[23], and initial studies drew a correlation between E-cadherin loss of expression and Wnt signaling activation in HCC[24]. These results led to the conclusion that E-cadherin exerts a dual tumor suppressor function in HCC: first by sequestering β-catenin away from Wnt signaling and second by maintaining cell–cell contact to prevent epithelial–mesenchymal transition (EMT) and metastasis[25]. However, this view may be too simplistic as E-cadherin expression in human HCC is largely variable and is elevated in 40% of HCC cases[26,27]. In addition, studies on other types of solid tumors have shown that E-cadherin can exert pro-tumorigenic functions[28,29]. These findings suggest that the role of E-cadherin in HCC is more complex than previously thought.

To elucidate the role of β-catenin during HCC progression, we have integrated patient data analysis with mechanistic studies using a pre-clinical model of HCC initiated upon functional inactivation of the Rb family, an almost universal event during HCC development[30,31]. We show that, despite the early occurrence of mutations targeting Wnt signaling components in human HCC, membrane localization of β-catenin is a dominant feature of HCC until the advanced stages of the disease when β-catenin progressively engages in Wnt signaling activation. At the plasma membrane, β-catenin interacts with multiple cadherin family members to promote the signaling of growth factor receptors such as epidermal growth factor receptor (EGFR) and support HCC cell survival. Overall, our study reveals the evolutive nature of β-catenin tumor-promoting functions during HCC progression.

## Results

**β-Catenin is restricted to the membrane until late-stage HCC.** Wnt pathway components are mutated in 30–40% of HCC patients[4]. However, when these mutations occur during the course of the disease and whether they directly translate into the activation of the Wnt signaling are not known. To determine the mutational status of Wnt signaling during HCC development, we took advantage of a The Cancer Genome Atlas (TCGA) HCC dataset representing patients from stages I to III. Analysis shows that mutations targeting Wnt signaling are present in ~35% of patients, independent of their tumor stage (Fig. 1a), indicating that these mutations represent an early event in HCC and their frequency remains stable thereafter. Next, we sought to address the subcellular localization of β-catenin during HCC progression and performed immunohistochemistry (IHC) for β-catenin in human HCC tissue arrays ranging from pre-malignant conditions to malignant stages (stages I–IV). β-Catenin is predominantly localized at the plasma membrane, but its abundance at the membrane progressively declines beyond stage I. Importantly, it is only at the stage IV that β-catenin is present in the nucleus

(54% of the samples) (Fig. 1b, c), which correlates with the late-stage activation of Wnt signaling reported in human HCC[32,33]. Therefore, we found a significant absence of correlation between Wnt signaling mutational status and β-catenin nuclear localization during HCC progression until stage IV (Fisher's test, p value <0.05). These results indicate that nuclear localization of β-catenin is not an immediate consequence of mutations targeting Wnt pathway components, suggesting that additional events are required for aberrant Wnt signaling activation.

**Membrane-localized β-catenin promotes HCC growth.** Localization of β-catenin at the membrane has been observed in multiple mouse models of HCC, either chemically induced by DEN or triggered by single or combined targeted genetic alterations (E2f1, Myc, Tgfβ, Akt, etc.)[17,34–36]. Therefore, human/mouse cross-species analyses show that β-catenin is commonly found at the cellular membrane in HCC.

To determine the mechanisms governing β-catenin localization and its functional consequence for HCC progression, we took advantage of a mouse model that we previously generated[31]. Rb-family proteins (Rb, p130, and p107) are key regulators of cell cycle activity and their functional alteration is almost universal during HCC initiation. Conditional inactivation of *Rb* gene family (triple knockout; TKO) in the adult mouse liver generates a well-differentiated type of HCC (TKO HCC) that recapitulates multiple features of the human disease[30,31]. Whereas β-catenin is expressed in different subcellular compartments of hepatocytes around the central vein (CV) in control liver (CTRL), it is exclusively detected at the membrane of tumor cells in TKO HCC (Fig. 2a), with an increase in β-catenin protein but not mRNA levels (Fig. 2b, c). Serially transplanted TKO HCC tumors exhibit a poorly differentiated morphology compared to the parental TKO HCC. IHC analysis for β-catenin localization showed that subcutaneous tumors display heterogeneous expression of nuclear and membranous β-catenin (Supplementary Fig. 1A–B). These results indicate that TKO HCC recapitulates the evolution of β-catenin localization observed in human HCC.

Immunofluorescence (IF) confirmed that β-catenin is expressed at the plasma membrane of cell lines previously derived from individual primary TKO HCC tumors (collectively designated as TKO HCC cells)[30,31] (Fig. 2d). Stable knockdown (KD) of β-catenin in TKO HCC cells by lentiviral-mediated expression of four independent hairpins (shβcat1-4) (Fig. 2e) identified shβcat3 and shβcat4 as the most effective in inhibiting β-catenin expression (Fig. 2f, g). In vitro growth assay showed that β-catenin KD alters TKO HCC cell morphology (Supplementary Fig. 2) and reduces their growth (Fig. 2h). Impaired cell growth following β-catenin KD is predominantly due to increased apoptosis rather than cell cycle arrest (Fig. 2i–k). In addition, subcutaneous transplantation assay showed that β-catenin

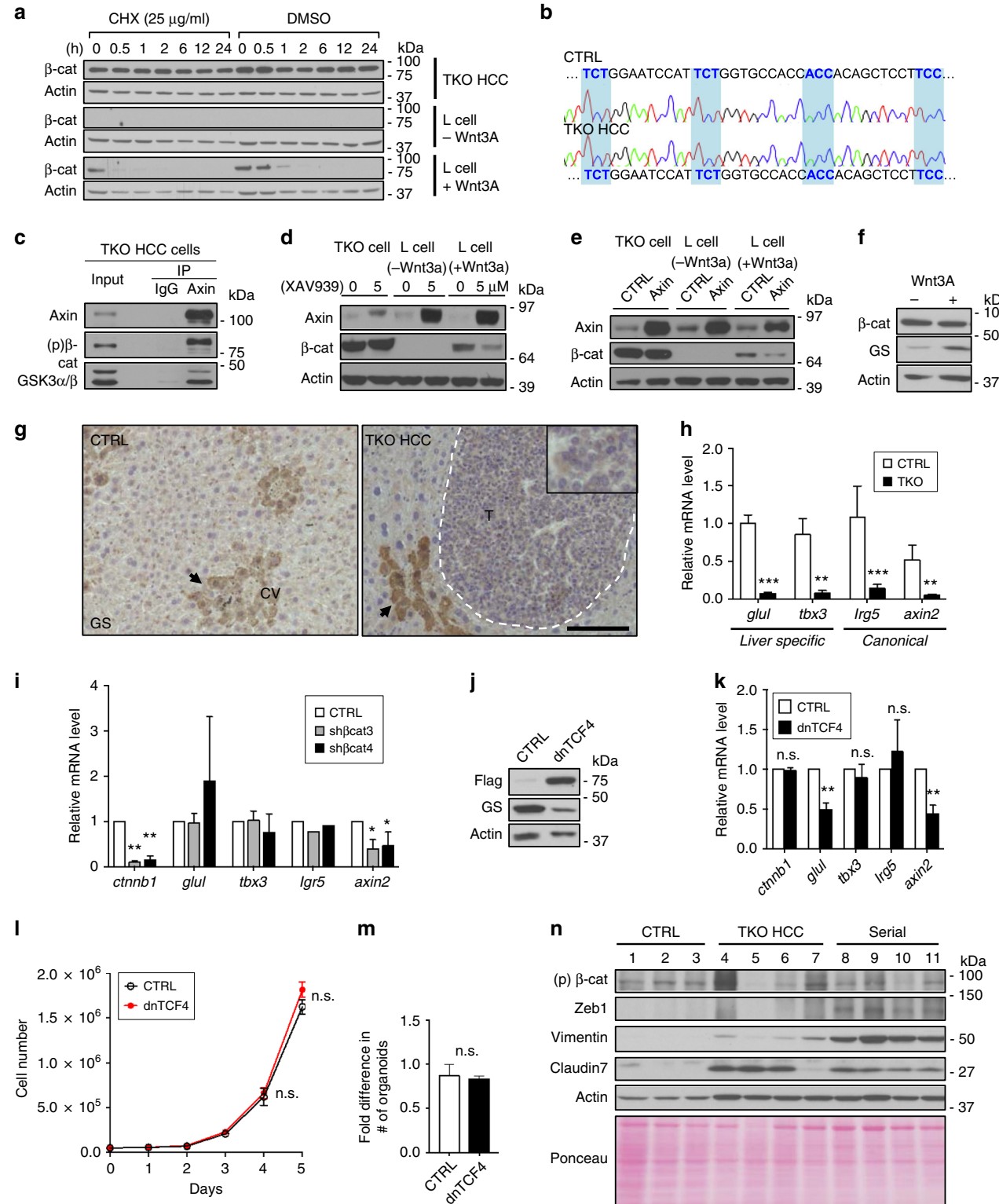

presence is required for the in vivo growth of TKO HCC cells (Fig. 2l). To further support the pro-survival role of β-catenin, we derived organoid culture from primary TKO HCC tumors[37]. After determining that β-catenin is also expressed at the membrane of organoid cells, we confirmed that silencing its expression by shβcat3 also decreased organoid-forming capacity (Fig. 2m, n). Collectively, these results demonstrate that β-catenin is localized at the membrane of TKO HCC and plays a critical role in the promotion of HCC cell growth.

**β-Catenin does not promote early HCC through Wnt signaling.** Minimal amount of β-catenin in the nucleus is sufficient to activate Wnt target genes[38]. Therefore, we cannot rule out the contribution of Wnt signaling in HCC cell survival even though β-catenin is undetectable in the nucleus of early-stage human HCC and TKO HCC cells by IHC. Protein translation inhibition using cycloheximide (CHX) revealed that β-catenin protein expression is stable in TKO HCC cells (Fig. 3a). Sequencing analysis revealed that *ctnnb1* is wild type in primary TKO HCC

**Fig. 3** β-Catenin does not promote early hepatocellular carcinoma (HCC) through Wnt signaling. **a** Immunoblot for β-catenin in triple knockout (TKO) HCC cells and Wnt3A-inducible mouse fibroblasts used as controls for non-active vs. active Wnt signaling (L cells; parental or Wnt3a+) treated with 25 μg/ml cycloheximide (CHX) or dimethyl sulfoxide (DMSO) for the time indicated (0–24 h). **b** Comparison of *ctnnb1* complementary DNA (cDNA) sequence in control liver (CTRL) and TKO HCC. Regions highlighted in blue correspond to the phosphorylation sites. **c** Immunoprecipitation (IP) of immunoglobulin G (IgG) and Axin in TKO HCC cells. The presence of Axin, glycogen synthase kinase 3β (GSK3α/β), and phospho-β-catenin in the pull-down fraction was determined by immunoblot. **d** TKO HCC cells and as L cells (parental or Wnt3a+) were treated with 5 μM XAV939 for 48 h. The expression levels of Axin and β-catenin were determined by immunoblot. **e** Axin cDNA was overexpressed in TKO HCC cells and L cells (parental or Wnt3a+). The expression levels of Axin and β-catenin were determined by immunoblot. **f** Immunoblot for β-catenin and glutamine synthetase (GS) in TKO HCC cells treated with Wnt3A media. **g** Immunohistochemistry (IHC) for GS in control liver (CTRL) and TKO HCC. Black arrowheads indicate central vein (CV). The tumor zone in TKO HCC is delineated by a white dotted line. Scale bar = 50 μm. **h** Quantitative reverse transcription PCR (RT-qPCR) analysis for *glul, tbx3, lgr5,* and *axin2* in control liver (CTRL; *n* = 4) and TKO HCC (*n* = 9). **i** RT-qPCR analysis for *glul, tbx3, lgr5,* and *axin2* in TKO HCC cells upon β-catenin knockdown (KD) with shβcat3-4 (*n* > 3). **j** Immunoblot for β-catenin and GS in TKO HCC cells overexpressing dominant-negative of TCF4 (dnTCF4). **k** RT-qPCR analysis for *glul, tbx3, lgr5,* and *axin2* in TKO HCC cells upon dnTCF4 overexpression (*n* = 3). **l** Growth curve of TKO HCC cells upon dnTCF4 overexpression (*n* = 2). **m** Fold difference in the number of TKO HCC organoids upon dnTCF4 overexpression (*n* = 2). **n** Immunoblot of phospho β-catenin (Ser33/37/Thr41) and epithelial–mesenchymal transition (EMT) markers (Zeb1, Vimentin, and Claudin7) in control (CTRL) livers (*n* = 3), primary TKO HCC tumors (*n* = 4) and serially transplanted TKO HCC (Serial) tumors (*n* = 4). Data are represented as mean ± SEM. *$p < 0.05$, **$p < 0.01$, and ***$p < 0.001$. n.s. not significant. See also Supplementary Fig. 3

and TKO HCC-derived cell lines (Fig. 3b and Supplementary Fig. 3A). In addition, immunoprecipitation (IP) assay showed that the destruction complex is intact (Fig. 3c and Supplementary Fig. 3B). These data suggest that β-catenin, although capable of being degraded by the destruction complex in TKO HCC, evades degradation via alternative means. Accordingly, treatment with XAV939[39], which stabilizes endogenous Axin[39], and ectopic Axin expression failed to significantly change β-catenin expression level (Fig. 3d, e). However, stimulation of TKO HCC cells with Wnt3a ligand increased glutamine synthetase (GS; a clinical marker for Wnt activity in the liver[40]) expression (Fig. 3f), indicating that Wnt pathway can be activated in TKO HCC.

To determine whether Wnt signaling is active in the context of membrane β-catenin localization, we first assessed the expression of key Wnt target genes in primary tumors. GS is expressed in the pericentral zone of normal liver but is undetectable in TKO HCC (Fig. 3g). In addition, quantitative PCR (qPCR) analysis showed a decreased expression of liver specific (*glul, tbx3*) and canonical (*lgr5, axin2*) Wnt target genes in TKO HCC compared to controls (Fig. 3h). However, silencing of β-catenin in TKO HCC cells only repressed the expression of *axin2*, but not *glul* and *tbx3* (Fig. 3i), suggesting that Wnt signaling is minimally active in TKO HCC.

To determine whether minimal Wnt signaling activity drives TKO HCC cell survival, we tested the transcriptional and functional consequences of suppressing Wnt target gene expression by overexpressing a dominant-negative form of TCF4 (dnTCF4)[41,42]. Ectopic expression of dnTCF4 in TKO HCC cells repressed the expression of *axin2* (similar to β-catenin silencing) and GS/*glul* (albeit this could be through a β-catenin-independent mechanism[43]) (Fig. 3j, k), but failed to repress TKO HCC cell and organoid proliferation (Fig. 3l, m). Collectively, these results suggest that, despite the presence of a functional Wnt pathway, Wnt signaling activity is minimal in TKO HCC and does not promote HCC cell survival.

Finally, we systematically tested the status of the key Wnt pathway components to elucidate the mechanism of progressive nuclear translocation of β-catenin upon TKO HCC serial transplantation. We found that *ctnnb1* is wild type and β-catenin is phosphorylated by the destruction complex in serially transplanted TKO HCC tumors (Supplementary Fig. 3C and Fig. 3n, top panel). In contrast, serially transplanted tumors displaying nuclear β-catenin expression had lost Axin1 expression (Supplementary Fig. 3D). Functionally, we found a correlation between the progression of TKO HCC and the onset of metastasis markers (Fig. 3n). Therefore, these data recapitulate the

observations made in advanced human HCC[22] and suggest that the activation of Wnt signaling in late-stage HCC plays a critical role in the initiation of metastasis.

**E-cadherin recruits β-catenin away from the Wnt pathway.** Increased β-catenin stability in the presence of a functional destruction complex suggests that it escapes degradation. Previous studies have demonstrated that the recruitment of β-catenin by the AJ complex results in delayed β-catenin protein turnover[11]. To test this possibility, we first confirmed co-localization of β-catenin and E-cadherin in TKO HCC cells (Fig. 4a) and in various human HCC cell lines (Supplementary Fig. 4A) by IF. IP assay showed that β-catenin interacts with both E-cadherin and α-catenin in TKO HCC and this interaction was not altered upon stimulation with Wnt3A ligand (Fig. 4b, c). In addition, we observed increased E-cadherin protein and *cdh1* mRNA expression in TKO HCC compared to the control liver, similar to what has been reported in human HCC[26] (Fig. 4d, e). *Cdh1* expression is predominantly regulated by a family of transcription factors involved in the EMT, such as Slug, Snail, and Twist[25], and we found decreased expression of *twist1* in TKO HCC (Fig. 4e). To determine the relevance of β-catenin recruitment by the AJ complex, we exploited TCGA HCC patient dataset and identified a correlation between β-catenin and E-cadherin protein (but not mRNA) expression levels in HCC patients, independent of the tumor stage (Fig. 4f). Despite having identical residue sequences, exogenous Flag-β-catenin had a shorter half-life than endogenous β-catenin (Supplementary Fig. 4B–C), suggesting that the recruitment of β-catenin by the AJ complex is not a post-translational mechanism. Collectively, these data suggest that increased E-cadherin expression promotes the recruitment of β-catenin to the plasma membrane by the AJ complex to prevent its degradation by the destruction complex in HCC.

To determine the consequence of β-catenin recruitment by the AJ complex for Wnt signaling in HCC, we used a set of established Wnt target genes (*glul, tbx3, lgr5,* and *axin2*) as a readout for Wnt signaling activity in the TCGA dataset of HCC patients and found that E-cadherin expression negatively correlates with Wnt signaling activity (Fig. 4g). Using a similar approach for TCGA datasets for three other gastro-intestinal tract cancers, we failed to observe a similar correlation (Fig. 4h). Collectively, these results suggest that the dominant recruitment of β-catenin within the AJ complex to the detriment of Wnt signaling is a unique feature of HCC.

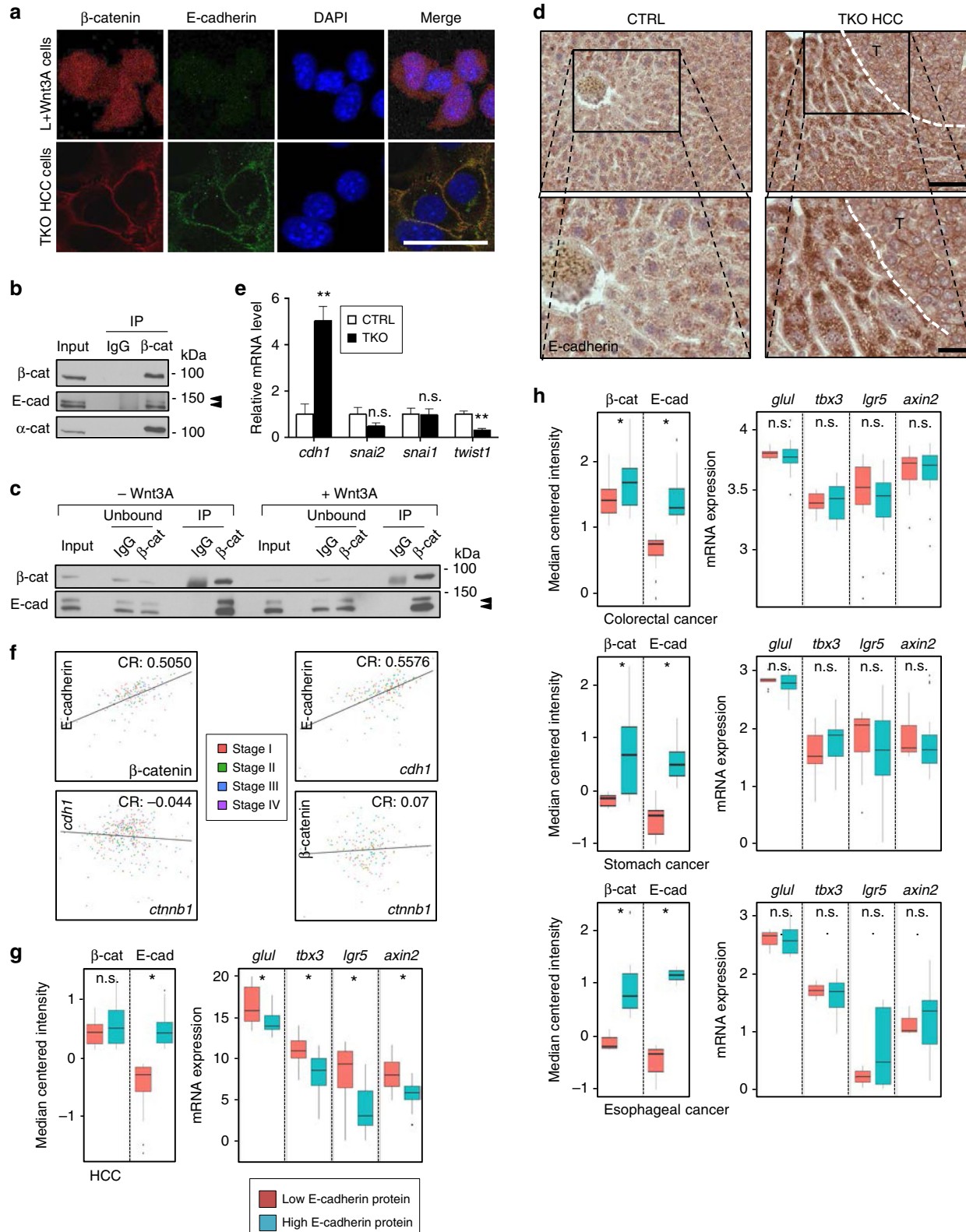

**Compound integration of β-catenin in the AJ complex in HCC.** To determine the role of the AJ complex in HCC, we silenced E-cadherin expression in TKO HCC cells using two independent hairpins (shEcad1–2). Efficient E-cadherin KD did not repress the proliferative capacity of TKO HCC cells, despite an alteration in cell morphology and cell–cell contact at low cell density (Fig. 5a and Supplementary Figu. 5A). IF analysis showed that β-catenin maintains membrane surface localization despite E-cadherin loss, suggesting a compensatory mechanism for E-cadherin loss of expression (Fig. 5b). To determine this compensatory mechanism, we analyzed β-catenin interactome in TKO HCC by IP followed by mass spectrometry (MS) and found that most β-catenin interactions occur with either the AJ complex or the degradation complex (Fig. 5c and Supplementary Table 1). This

**Fig. 4** E-cadherin recruits β-catenin away from the Wnt pathway. **a** Immunofluorescence (IF) for β-catenin (red), E-cadherin (green), and 4′,6-diamidino-2-phenylindole (DAPI) (blue) in triple knockout hepatocellular carcinoma (TKO HCC) and Wnt3a+ L cell (a positive control for nuclear β-catenin). Scale bar = 25 μm. **b** Immunoprecipitation (IP) for immunoglobulin G (IgG) and β-catenin in TKO HCC cells. The presence of β-catenin, E-cadherin, and α-catenin were detected in the pull-down fraction by immunoblot. **c** IP for IgG and β-catenin in TKO HCC cells treated with Wnt3A media. The presence of β-catenin and E-cadherin were detected in both unbound and pull-down fraction by immunoblot. Double arrowheads indicate two migrating forms of E-cadherin. **d** Immunohistochemistry (IHC) for E-cadherin in control liver (CTRL) and TKO HCC. The tumor zone in TKO HCC is delineated by a white dotted line. Scale bar = 100 μm (top), 25 μm (bottom). **e** Quantitative reverse transcription PCR (RT-qPCR) analysis in control livers (CTRL, *n* = 4) and TKO HCC (*n* = 8) for *cdh1*, *snai2*, *snai1*, and *twist*. **f** Correlation between E-cadherin vs. β-catenin (left top; *p* value = 2.509e−12), *cdh1* vs. *ctnnb1* (left bottom; *p* value = 4.861e−3), *cdh1* vs. E-cadherin (right top; *p* value = 2.426e−16) and *ctnnb1* vs. β-catenin (right bottom; *p* value = 0.691) expressions in HCC patients at different stages of the disease (CR = correlation ratio; Pearson's product-moment correlation). **g, h** Analysis of protein and mRNA expression from The Cancer Genome Atlas (TCGA) patient dataset for **g** HCC and **h** other gastro-intestinal cancers (colorectal, stomach, and esophageal). Samples were separated based on E-cadherin protein expression (low vs. high). Expression of *glul*, *tbx3*, *lgr5*, and *axin2* were assessed between these two groups to determine the correlation (non-parametric *t* test) between E-cadherin expression and Wnt target gene expressions in different cancers. Data are represented as mean ± SEM. *\*p* < 0.05 and *\*\*p* < 0.01. n.s. not significant. See also Supplementary Fig. 4

unbiased approach showed that, in addition to E-cadherin, β-catenin also interacts with N- and K-cadherin, which was confirmed by IP (Fig. 5d). We failed to detect by IP the interaction between β-catenin and K-cadherin upon N-cadherin KD (shNcad1), which suggests that N-cadherin may mediate β-catenin/K-cadherin binding. Next, we performed E/N-cadherin single KD or double KD (DKD) in TKO HCC cells and found that silencing of both cadherins leads to reduced half-life and decreased expression of β-catenin (Fig. 5e, f and Supplementary Fig. 5B), without any effect on Wnt target gene (*glul*, *tbx3*, and *axin2*) expression (Supplementary Fig. 5C). Combined E- and N-cadherin repression led to decreased TKO HCC cell growth in vitro with altered cell morphology and impaired cell–cell contact (Fig. 5g and Supplementary Fig. 5D). Of note, N-cadherin expression did not increase in TKO HCC compared to controls, suggesting a mechanism specific for E-cadherin up-regulation in TKO HCC (Supplementary Fig. 5E–F).

These data indicate that compound integration of β-catenin by several members of the cadherin family in the AJ complex is crucial for β-catenin retention at the membrane and HCC cell survival.

To determine the long-term consequences of disrupting the AJ complex in HCC development, TKO HCC cells with E/N-cadherin single KD or DKD were subcutaneously transplanted into the flank of immunocompromised nonobese diabetic/severe combined immunodeficiency (nod-scid) mice. In contrast to the growth inhibition observed in an in vitro short-term context, DKD did not impair tumor formation in vivo (Fig. 5h) and actually lead to increased cell cycle activity (Supplementary Fig. 5G). Histological and IHC analyses revealed that individual clones lacking detectable β-catenin expression progressively emerge from double E/N-cadherin-deficient tumor masses (Fig. 5i and Supplementary Fig. 5H). While there was no difference in tumor grade among the different conditions, cells within the E/N-cadherin DKD tumors displayed heterogeneous nuclear size and ploidy (Fig. 5j and Supplementary Fig. 5I–J). In addition, IHC analysis revealed a mosaic GS expression in the E/N-cadherin DKD tumors, indicative of heterogeneous Wnt signaling activity (Fig. 5k). Interestingly, we could not find co-occurrence of increased nuclear size and GS expression, suggesting that Wnt signaling activation and DNA content alteration are mutually exclusive, in line with previous observations[16,34]. Collectively, these data suggest that the long-term consequences of disrupting the AJ complex, which include increased proliferation, heterogeneous nuclear size, and ploidy, as well as Wnt signaling activation, are beneficial for tumor growth.

**EGFR signaling promotes HCC cell survival.** To determine the mechanism by which the AJ complex supports HCC cell survival,

we systematically investigated the consequences of targeting each of its established functions (cytoskeleton maintenance, cell–cell interaction, and promotion of growth factor signaling) in TKO HCC. We first disrupted AJ complex interaction with the intracellular cytoskeleton by repressing the expression of α-catenin, which acts as a bridge between the AJ complex and F-actin filaments[8]. α-Catenin KD altered the morphology (but not the cell–cell contact) of TKO HCC cells (Supplementary Fig. 6A), but did not impact cell growth nor β-catenin expression (Fig. 6a), indicating that the AJ complex does not promote HCC cell survival by preserving the integrity of the cytoskeleton. We next disrupted the cell–cell adhesion activity of the AJ complex by overexpressing a dominant-negative form of E-cadherin (DNE) subcloned into MigR1-GFP (co-expressing the GFP protein as a selection marker)[44]. DNE lacks most of the ectodomain of E-cadherin (required for the maintenance of cell–cell adhesion), but retains the transmembrane domain (for anchoring E-cadherin to the plasma membrane) and the cytoplasmic domain (Supplementary Fig. 6B), allowing DNE to compete with endogenous E-cadherin, and in particular its slow migrating form, for β-catenin binding (Fig. 6b, c). IF showed that DNE expression increases β-catenin expression and re-directs its subcellular localization in TKO HCC and in human HCC cells (Fig. 6d and Supplementary Fig. 6C–D). DNE expression did not impair β-catenin half-life or TKO HCC growth in vitro (Fig. 6e and Supplementary Fig. 6E) and wound healing assays showed no difference in the migratory capacity of TKO HCC cells upon DNE expression (Supplementary Fig. 6F). DNE-expressing TKO HCC cells were subfractionated based on their GFP expression, which showed a correlation between DNE expression and the disrupted cell–cell adhesion without any consequences on their growth (Supplementary Fig. 6F-G). Collectively, these data indicate that β-catenin within the AJ complex supports HCC cell survival by mechanisms independent of cytoskeletal maintenance or cell–cell adhesion.

Finally, cadherins also interact with several receptor tyrosine kinases (RTKs), with varying and sometimes conflicting consequences[45]. Among these RTKs, E-cadherin interacts with EGFR through its ectodomain[46]. EGFR is overexpressed in 40–70% of HCC and treatment of HCC cell lines, xenograft, and mouse models with EGFR inhibitors represses cell growth[47,48], indicating that EGFR signaling is critical for HCC cell proliferation. To determine whether EGFR signaling supports TKO HCC cell survival, we first silenced EGFR expression by short hairpin RNA (shRNA). Repression of EGFR expression decreases TKO HCC cell growth in a dose-dependent manner. Analysis of other EGFR family members shows no compensation by EGFR2, while EGFR3 was undetectable by qPCR and immunoblot analysis (Fig. 6f). Next, we tested the capacity of EGFR inhibitors to

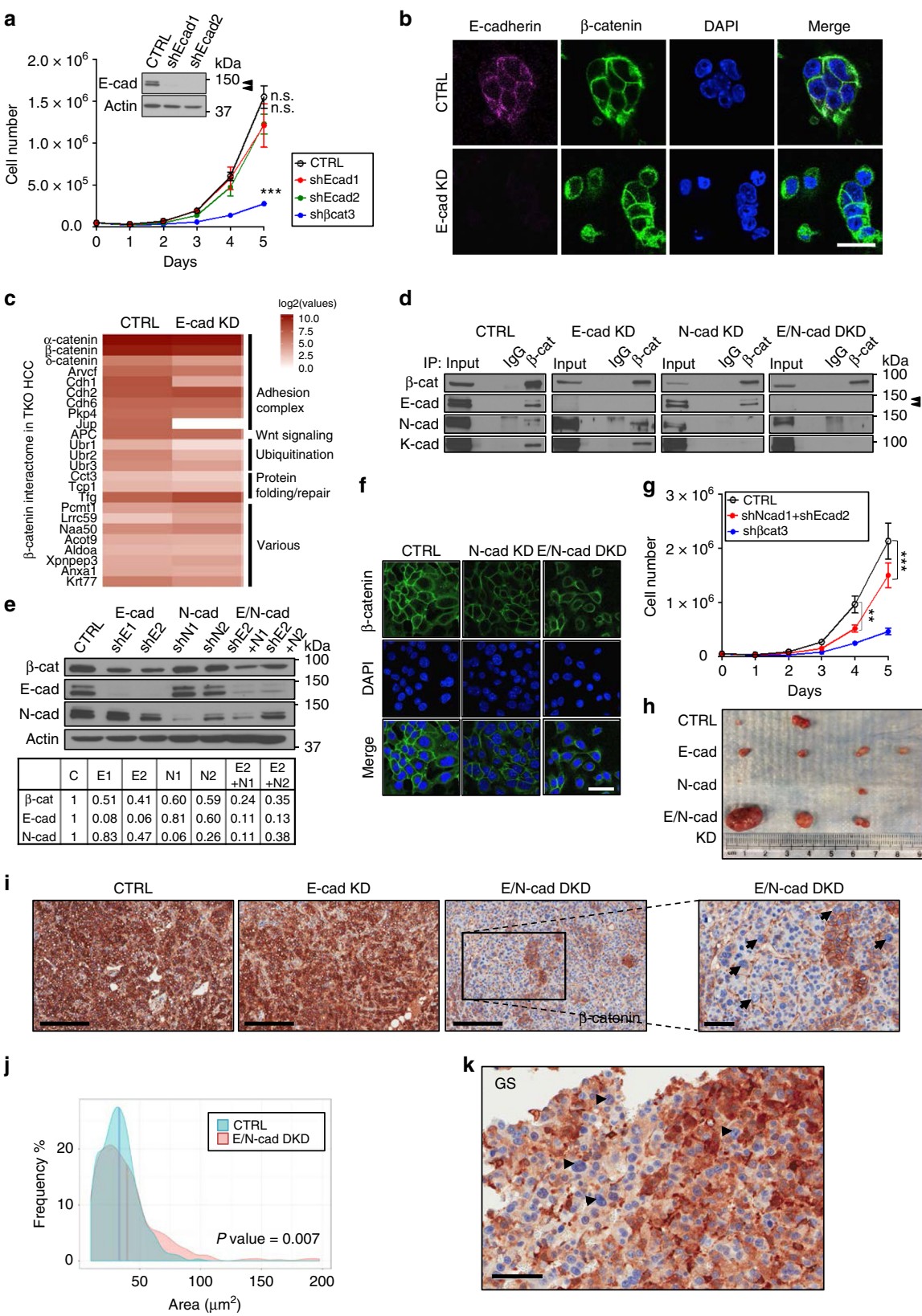

repress TKO HCC survival. We treated TKO HCC cells and organoids derived from primary TKO HCC tumors with Erlotinib® (EGFR inhibitor[49]), Afatinib® (EGFR family inhibitor[50]), and Sorafenib® (a multi-kinase inhibitor used as a control[51]). In these conditions, we found that EGFR inhibition by both Erlotinib and Afatinib trigger apoptosis of TKO HCC

cells and TKO HCC organoids (Fig. 6g, h). Interestingly, significant effects were already observed at 1 μM Afatinib vs. 10 μM Erlotinib, suggesting that targeting multiple EGFR family members may be more potent to trigger TKO HCC cell death. Collectively, these results show that EGFR inhibition impairs TKO HCC growth.

**Fig. 5** Compound integration of β-catenin in the adherens junction (AJ) complex in hepatocellular carcinoma (HCC). **a** Growth curve of triple knockout HCC (TKO HCC) cells upon E-cadherin knockdown (KD) with shEcad1–2 ($n = 4$). E-cadherin protein level is determined by immunoblot. Double arrowheads indicate two migrating forms of E-cadherin. **b** Immunofluorescence (IF) for E-cadherin (magenta), β-catenin (green), and 4′,6-diamidino-2-phenylindole (DAPI (blue)) in TKO HCC cells expressing an empty vector (control (CTRL)) or shEcad2 (E-cadherin KD). Scale bar = 25 μm. **c** Mass spectrometry (MS) analysis identifies the interactome of β-catenin in TKO HCC cells upon E-cadherin KD (shEcad2), relative to the control (CTRL). **d** Immunoprecipitation (IP) of immunoglobulin G (IgG) and β-catenin in TKO cells expressing an empty vector (CTRL), shEcad2 (E-cadherin KD), shNcad1 (N-cadherin KD), or shEcad2 + shNcad1 (E/N-cadherin double knockdown (DKD)). Endogenous β-catenin was pulled down and β-catenin, E-cadherin, N-catenin, and K-cadherin were detected by immunoblot. Double arrowheads indicate two migrating forms of E-cadherin. **e** β-Catenin, E-cadherin and N-cadherin expression in TKO HCC cells expressing an empty vector (CTRL), shEcad2 (E-cadherin KD), shNcad1 (N-cadherin KD), or shEcad2 + shNcad1 (E/N-cadherin DKD). Values in the lower panel represent the intensity of corresponding bands, normalized by actin. **f** IF for β-catenin (green) and DAPI (blue) in TKO HCC cells expressing an empty vector (CTRL), shNcad1 (N-cadherin KD), or shEcad2 + shNcad1 (E/N-cadherin DKD). **g** Growth curve of TKO HCC cells expressing an empty vector (CTRL) or shEcad2 + shNcad1 (E/N-cadherin DKD). ($n = 3$). Scale bar = 25 μm. **h** Subcutaneous tumors from nonobese diabetic/severe combined immunodeficiency (nod/scid) mice injected with TKO HCC cells expressing an empty vector (CTRL), shEcad2 (E-cadherin KD), shNcad1 (N-cadherin KD), or shEcad2 + shNcad1 (E/N-cadherin DKD). **i** Immunohistochemistry (IHC) for β-catenin in subcutaneous tumors. **j** Quantification of nuclei size in subcutaneous tumors from TKO HCC cells, control (CTRL) or E/N-cadherin DKD ($n = 3$). Scale bar = 200 μm, 60 μm (enlarged). **k** IHC for glutamine synthetase (GS) in subcutaneous tumor from TKO HCC cells with E/N-cadherin DKD. Cells with enlarged nuclei (arrowheads) do not express GS, indicating that Wnt activity is not required to trigger genomic instability. Scale bar = 50 μm. Data are represented as mean ± SEM. **p < 0.01 and ***p < 0.001. n.s. not significant. See also Supplementary Fig. 5

**The AJ complex enhances EGFR signaling**. To elucidate the potential regulation of EGFR signaling by the AJ complex in TKO HCC, we first determined whether the components of the AJ complex interact with EGFR. IP assay showed that EGFR interacts with both E- and N-cadherin but not with β-catenin (Fig. 7a), suggesting an indirect interaction between EGFR and β-catenin. While β-catenin interacts with both migrating forms of E-cadherins, we observed that EGFR only interacts with the slow migrating form of E-cadherin, suggesting two pools of E-cadherin/β-catenin complex in the cell. Silencing E-cadherin repressed EGFR expression (Fig. 7b). In addition, silencing β-catenin expression led to a significant decrease in E-cadherin and EGFR expression and delayed EGFR activation upon EGF stimulation (Fig. 7c). Repression of EGFR expression upon β-catenin silencing appears to be due to post-translational mechanism since *egfr* mRNA was not altered upon silencing β-catenin (Fig. 7d). Interestingly, although silencing β-catenin repressed the expression of both E-cadherin and EGFR, it did not impair EGFR–E-cadherin interaction (Fig. 7e). Phosphorylation of β-catenin by multiple tyrosine kinases on residue Tyr654 disrupts β-catenin interaction with E-cadherin[52]. Treatment of TKO HCC cells with orthovanadate salt, a general phosphotyrosyl phosphatase inhibitor[53], disrupted the interaction between β-catenin and the slow migrating form of E-cadherin and decreased the expression of β-catenin, E-cadherin, and EGFR (as illustrated by the difference of signal intensity in the input lanes) (Fig. 7f). Overall, our results from genetic (shRNA silencing of AJ complex components) and pharmacological disruption of the β-catenin/cadherins interaction indicate that the AJ complex interacts with EGFR to promote the receptor stability and enhance EGFR signaling. Analysis of Tyr654 phosphorylation status shows decreased phosphorylation in TKO HCC (two out of nine) compared to controls (five out of six) (Supplementary Fig. 7A). However, pTyr654 β-catenin was undetectable in serially transplanted tumors, suggesting that different post-translational modifications[11] may regulate AJ complex stability during TKO HCC evolution.

Analysis of the control liver, and primary and serially transplanted TKO HCC samples indicates that EGFR expression decreases in TKO HCC vs. control livers (likely as a consequence of decreased mRNA abundance; Supplementary Fig. 7B) and its activation (phosphorylation of the Tyr1068 residue as a readout) becomes heterogeneous. However, EGFR and activated EGFR become undetectable in subcutaneous tumors characterized by a progressive shift in β-catenin localization, further supporting the

concept that an intact AJ complex is required for EGFR stability and signaling (Fig. 7g). Interestingly, we also observed a similar expression pattern for EGFR2, further suggesting that the AJ complex may stabilize multiple growth factor receptors in HCC.

Upon its stimulation, EGFR is internalized to be either degraded through the ubiquitin/lysosome pathway or to be recycled back to the membrane[54]. To determine the fate of EGFR upon disruption of its stabilizing interaction with the AJ complex, we treated TKO HCC cells with CHX for increasing amount of time and found that repression of β-catenin increased the degradation of EGFR, suggesting that EGFR is degraded and not recycled upon disruption of the AJ complex (Fig. 7h). Accordingly, exogenous expression of Flag-Ubiquitin followed by the IP of the Flag peptide showed an increased presence of EGFR in the pull-down fraction upon β-catenin silencing (Fig. 7i). Similarly, IP of EGFR showed an increased presence of ubiquitin chains upon silencing of β-catenin (Fig. 7j). Collectively, these results indicate that disruption of the interaction between EGFR and the AJ complex promotes EGFR degradation by the ubiquitin/lysozyme pathway.

**Correlation between E-cadherin and EGFR in HCC patients**. To investigate the relationship between EGFR signaling and AJ complex in human HCC, we interrogated the TCGA dataset. Analysis of EGFR and AJ complex protein expression in human HCC identifies a significant correlation between total EGFR and either E-cadherin or β-catenin expression (Fig. 8a). A positive correlation was also observed between activated EGFR and E-cadherin, but failed to reach significance. When repeated on a gender basis, we found a significant correlation within the male, but not the female group. Further analysis of the female group identified a subgroup of outlier patients with low EGFR expression (denoted by black dots) who display decreased expression of the EGF ligand (fold repression: 3.45 $^{\log 2}$, $p$ value < 0.004; DESeq2 Analysis) (Supplementary Data File), and exclusion of these outlier patients restores a significant correlation between activated EGFR and E-cadherin within the female group (Fig. 8b). Although the mechanism underlying the gender exclusive decreased EGFR/EGF expression is not known, these results establish a correlation between EGFR expression/signaling activity and AJ complex abundance in human HCC. In addition, detection of EGFR and E-cadherin expression in multiple epithelial cancers reveals a significant correlation for the following types of cancers (head and neck, colorectal cancer, and non-small-cell lung carcinoma (NSCLC)) out of seven tested

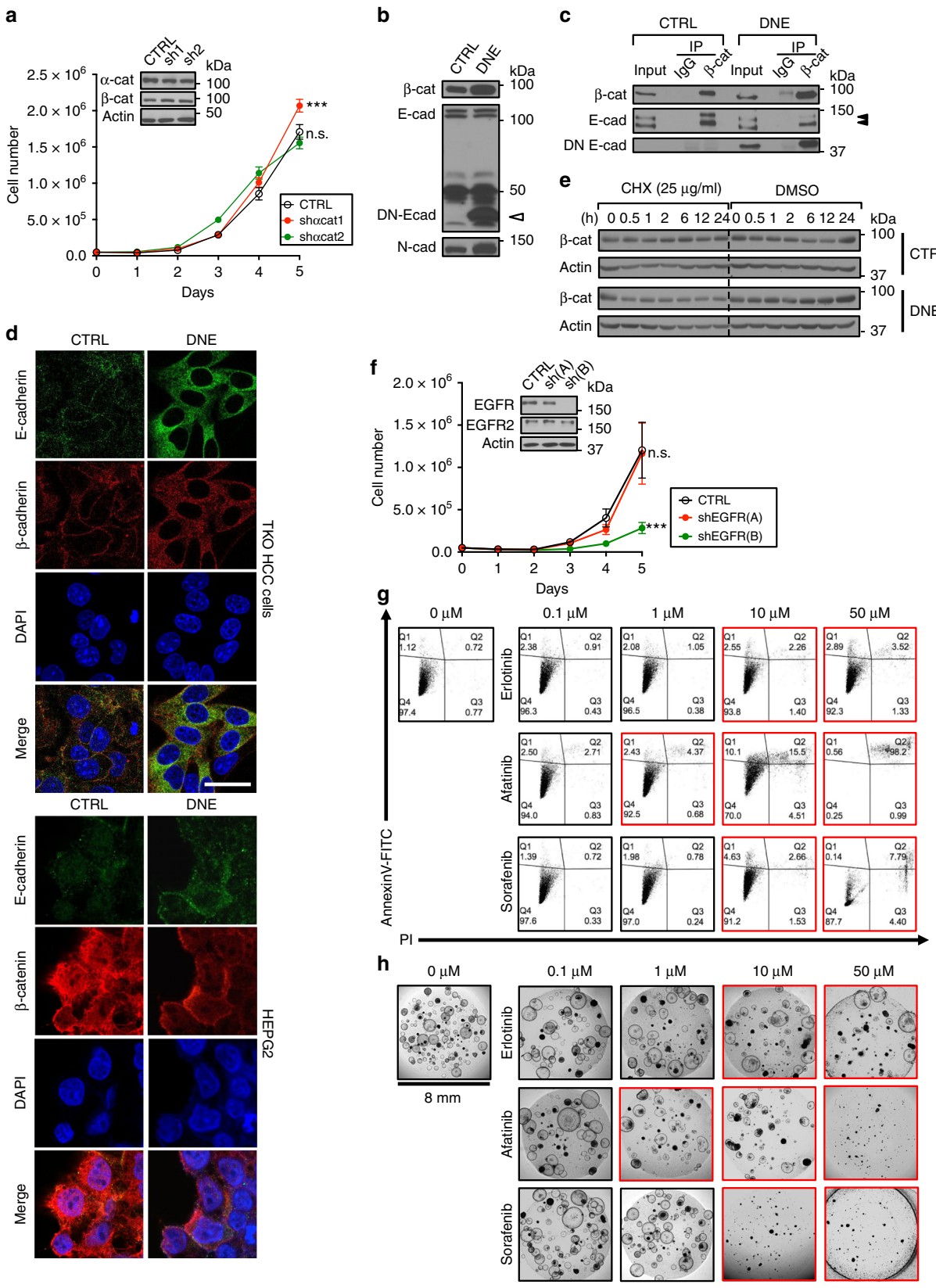

(Supplementary Fig. 8A). These cancers display mutations of the *egfr* gene in 4–14% of the cases, suggesting that AJ complex-mediated stabilization and genetic alteration may not represent mutually exclusive mechanisms to promote EGF signaling in cancer (Supplementary Fig. 8B). Finally, the increased apoptotic

activity of Afatinib vs. Erlotinib and the correlation between EGFR2 expression and β-catenin presence at the membrane during TKO HCC progression suggest that the AJ complex may expand its stability promoting functions to other EGFR family members. Accordingly, analysis of human HCC also indicates a

**Fig. 6** Epidermal growth factor receptor (EGFR) signaling promotes hepatocellular carcinoma (HCC) cell survival. **a** Growth curve of triple knockout (TKO) HCC cells upon α-catenin KD with shαcat1–2 (n = 6). α-Catenin and β-catenin protein levels upon expression of shαcat1–2, as detected by immunoblot. **b** Immunoblot for E-cadherin and N-cadherin in TKO HCC cells infected with empty MigR1 (control (CTRL)) or MigR1 dominant-negative form of E-cadherin (DNE). White arrowhead indicates DN-Ecad. **c** Immunoprecipitation (IP) of immonoglobulin G (IgG) and β-catenin in TKO HCC cells expressing control (CTRL) or DNE. The expressions of β-catenin, E-cadherin, and DNE were detected by immunoblot. Double arrowheads indicate two migrating forms of E-cadherin. **d** IF for E-cadherin (green), β-catenin (red), and 4′,6-diamidino-2-phenylindole (DAPI (blue) in TKO HCC cells and HEPG2 cells expressing control (CTRL) or DNE. Scale bar = 25 μm. **e** Immunoblot for β-catenin in TKO HCC cells expressing control (CTRL) or DNE. Cells were treated with 25 μg/ml cycloheximide (CHX) for the time indicated (0–24 h). **f** Growth curve of TKO HCC cells upon EGFR KD with shEGFR(A–B) (n = 3). EGFR and EGFR2 (HER2) protein levels upon expression of shEGFR(A–B), as detected by immunoblot. **g** Representative fluorescence-activated cell sorting (FACS) plot for AnnexinV-FITC expression on TKO HCC cells upon 0–50 μM Afatinib, Erlotinib, and Sorafenib treatment for 24 h. Dosage conditions that have significant effect on cells are highlighted in red. **h** Representative phase contrast images of TKO HCC organoids upon 0–50 μM Afatinib, Erlotinib, and Sorafenib treatment for 10 days. Dosage conditions that have significant effect on organoids are highlighted in red. Data are represented as mean ± SEM. ***p < 0.001. n.s. not significant. See also Supplementary Fig. 6

strong correlation between the expression of EGFR2 and EGFR3 and E-cadherin (Fig. 8c). Collectively, our results identify the promotion of epidermal growth factor signaling through cadherin maintenance as a critical tumor-promoting function of membranous β-catenin in HCC.

## Discussion

In this study, we provide evidence that β-catenin in the AJ complex supports HCC cell survival by promoting EGFR signaling during the earlier stages of HCC, and that its Wnt signaling associated role is restricted to late-stage HCC. Our report challenges the conventional view that β-catenin is a dominant oncogenic driver and instead highlights β-catenin as a tumor promoter with evolving functions during HCC development (Fig. 8d).

The AJ complex is best known to regulate cell–cell interaction and maintain F-actin-composed cytoskeleton, but here we show that the AJ complex also supports HCC through EGFR signaling. E-cadherin binds to EGFR in various solid tumors to modulate its activation but the pro- and anti-tumorigenic consequences appear to be context-dependent[45,55]. In the context of HCC, our data indicate that cadherins stabilize EGFR, and AJ complex disruption impairs EGFR stability, signaling, and tumor cell survival. EGFR signaling is often hyperactive in HCC, and its inhibition is detrimental for HCC cell survival[47,56]. However, EGFR inhibition in HCC does not impair tumor progression as the significant level of acute tumor cell death associated with EGFR inhibition induces compensatory proliferation and fosters tumor evolution[57]. Supportive of this, DEN- and DEN/Phenobarbital-induced HCC develop faster in the context of β-catenin deficiency, suggesting a paradoxical role of β-catenin in HCC[20,21]. In these experimental models, β-catenin is located at the plasma membrane and its deficiency impairs EGFR expression. Therefore, our data elucidate the mechanism of paradoxical tumor progression upon β-catenin deficiency/inhibition by establishing the connection between the AJ complex and EGFR signaling in HCC.

The fact that mutations targeting β-catenin, Axin, and APC occur as early as stage I and their frequency remains constant throughout HCC development suggests that Wnt signaling may nevertheless exert a pro-tumorigenic function at the early stage of HCC development. Our data determine that β-catenin-driven promotion of cell survival is independent of Wnt signaling, but limited data suggest that Wnt signaling could support more discrete function(s) such as escaping immune surveillance[58]. Despite the early occurrence of Wnt-activating mutations, it is not until stage IV that we observe a drastic shift in β-catenin localization, indicating that β-catenin localization is not solely dependent on the mutational status of the Wnt pathway. The mechanisms that ultimately trigger β-catenin cellular redistribution in advanced HCC remain unresolved, although the effect of orthovanadate treatment on β-catenin/E-cadherin interaction suggests that

increased tyrosine phosphorylation represents an important mechanism to disrupt AJ complex stability.

Our data indicate that AJ complex disruption has multiple beneficial consequences for advanced HCC progression. Although AJ complex disruption is detrimental for tumor growth in the short-term (as illustrated by decreased cell growth in vitro), long-term consequences are multiple and appear to be mutually exclusive such as altered DNA content and Wnt signaling activation. In contrast to colorectal cancer and hepatoblastoma, where Wnt signaling exerts a well-established driver role[12–14], the functional consequences of aberrant Wnt signaling for HCC remain obscure, as illustrated by its association with either good or bad prognosis[59]. Our data, as well as others, indicate that aberrant Wnt signaling is a feature of advanced HCC, characterized by the activation of multiple other oncogenic pathways (PI3k, Myc, Tgfβ, etc.)[22]. This molecular context raises the question of the exact contribution of Wnt signaling for late-stage HCC maintenance and progression. In particular, the mosaic expression of GS (Fig. 5k), a key marker of Wnt signaling in HCC[60], supports the concept that aberrant Wnt signaling is not a dominant oncogene in late-stage HCC. Based on Fig. 3n and previous reports[22], we propose that activation of Wnt signaling in advanced-stage HCC promotes specific features such as the initiation of metastasis.

In conclusion, our results challenge the current view on Wnt/β-catenin signaling in HCC and reveal an unexpected role for β-catenin in HCC development, with emphasis on its complex and evolving tumor-promoting function during the course of the disease. Future research will specifically focus on targeting AJ complex-associated tumor-promoting functions of β-catenin to impact HCC development.

## Methods

**Mice.** Rb-family cTKO mice were maintained on a mixed 129Sv/J;C57/BL6 background[31]. All experiments with mice were approved by the Children's Hospital of Philadelphia Institutional Animal Care and Use Committee (CHOP IACUC) (protocol #969). All animal studies were conducted in compliance with the ethical regulations for animal testing and research. For intrasplenic injection, mice were anesthetized and surgically opened on the upper left flank to inject adenovirus into the spleen. For subcutaneous serial transplantation, independent HCC primary tumors from the parental TKO mice were dissociated into single cells. The tumor cells (ranging $1 \times 10^3$–$10^6$) were resuspended in Matrigel and injected subcutaneously into the flanks of nod/scid mice. The resulting subcutaneous tumors [Generation 1] were harvested, dissociated, and subcutaneously transplanted [Generation 2]. For subcutaneous TKO HCC cell injection, TKO cells (ranging $5 \times 10^4$–$10^5$) were sorted and injected subcutaneously into nod/scid mice. Experiment endpoint was determined by the first appearance of tumors >1 cm².

**Cell culture.** Rb-family TKO 1.1 and 2.1 cell lines were derived from independent TKO HCC tumors[30]. Experiments have been performed in both cell lines with similar results and they are collectively referred to as TKO HCC cells. TKO HCC, HepG2 (ATCC® HB-8065), Hep3B (ATCC® HB-8064), and Huh-7 (JCRB0403 HuH-7) cells were cultured in DMEM (Mediatech), supplemented with 10% fetal bovine serum (FBS)/1% penicillin–streptomycin–glutamine (Invitrogen). SNU449 (ATCC® CRL-2234) cells were cultured in RPMI, supplemented with 10% FBS/1%

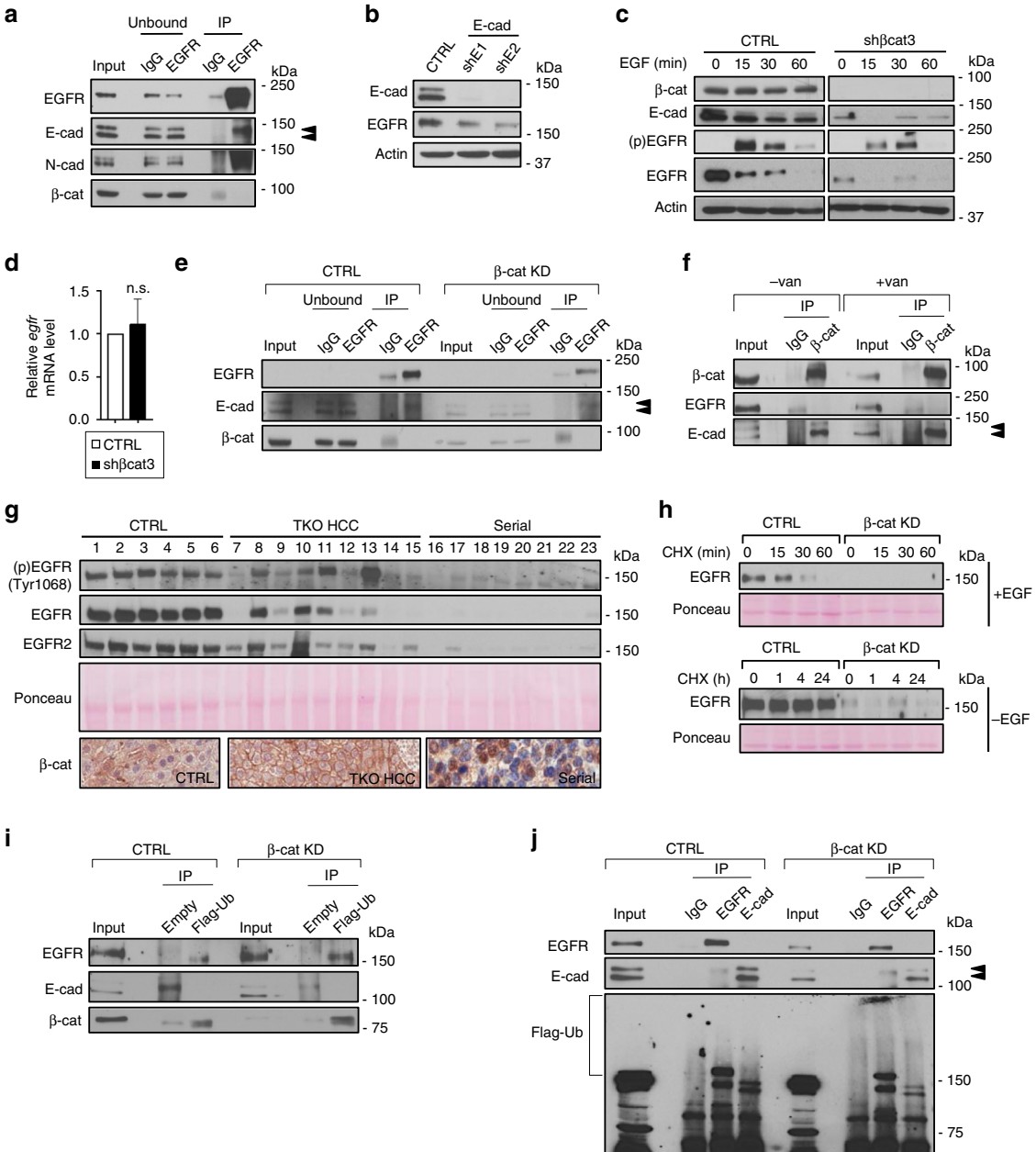

**Fig. 7** The adherens junction (AJ) complex enhances epidermal growth factor receptor (EGFR) signaling. **a** Immunoprecipitation (IP) for immunoglobulin G (IgG) and EGFR in triple knockout hepatocellular carcinoma (TKO HCC) cells. The presence of EGFR, E-cadherin, N-cadherin, and β-catenin were detected in both unbound and pull-down fractions by immunoblot. Double arrowheads indicate two migrating forms of E-cadherin. **b** Immunoblot for E-cadherin and EGFR in TKO HCC cells expressing control (CTRL) or shEcad1–2 (E-cadherin KD). **c** Serum-starved TKO HCC cells expressing control (CTRL) or shβcat3 (β-catenin KD) were treated with EGF (20 ng/ml) for the time indicated (0–60 min). β-Catenin, E-cadherin, P-EGFR(Tyr1068, activated EGFR), and EGFR were detected by immunoblot. **d** Quantitative reverse transcription PCR (RT-qPCR) analysis for *egfr* mRNA levels in TKO HCC cells expressing control (CTRL) or shβcat3 (β-cadherin KD). **e** IP for IgG and EGFR in TKO HCC cells expressing control (CTRL) or shβcat3 (β-cadherin KD). EGFR, E-cadherin, and β-catenin were detected in both unbound and pull-down fractions by immunoblot. Double arrowheads indicate two migrating forms of E-cadherin. **f** IP of IgG and β-catenin in TKO HCC cells treated with 0.5 mM orthovanadate. β-catenin, EGFR, and E-cadherin were detected by immunoblot. Double arrowheads indicate two migrating forms of E-cadherin. **g** P-EGFR (Tyr1068, activated EGFR), EGFR, and EGFR2 were detected in control livers (1–6), TKO HCC tumors (7–15), and serially transplanted TKO HCC (Serial) tumors (16–23) by immunoblot. β-Catenin immunohistochemistry (IHC) is provided as a reference for β-catenin subcellular localization. Ponceau serves as a loading control. **h** Immunoblot for EGFR in TKO HCC cells expressing control (CTRL) or shβcat3 (β-catenin KD). The cells were treated with 25 μg/ml cycloheximide (CHX) in the presence (upper panel) or absence (lower panel) of 20 ng/ml EGF for the time indicated (upper: 15–60 min; lower: 0–24 h). Ponceau serves as a loading control. **i** IP of Flag epitope in TKO HCC cells expressing control (CTRL) or shβcat3 (β-cadherin KD) as well as ectopic Flag-Ubiquitin. EGFR, E-cadherin, and β-catenin were detected by immunoblot. **j** IP of EGFR and E-cadherin in TKO HCC cells expressing control (CTRL) or shβcat3 (β-cadherin KD), as well as ectopic Flag-Ubiquitin. EGFR, E-cadherin, and Flag were detected by immunoblot. Double arrowheads indicate two migrating forms of E-cadherin. n.s. not significant. See also Supplementary Fig. 7

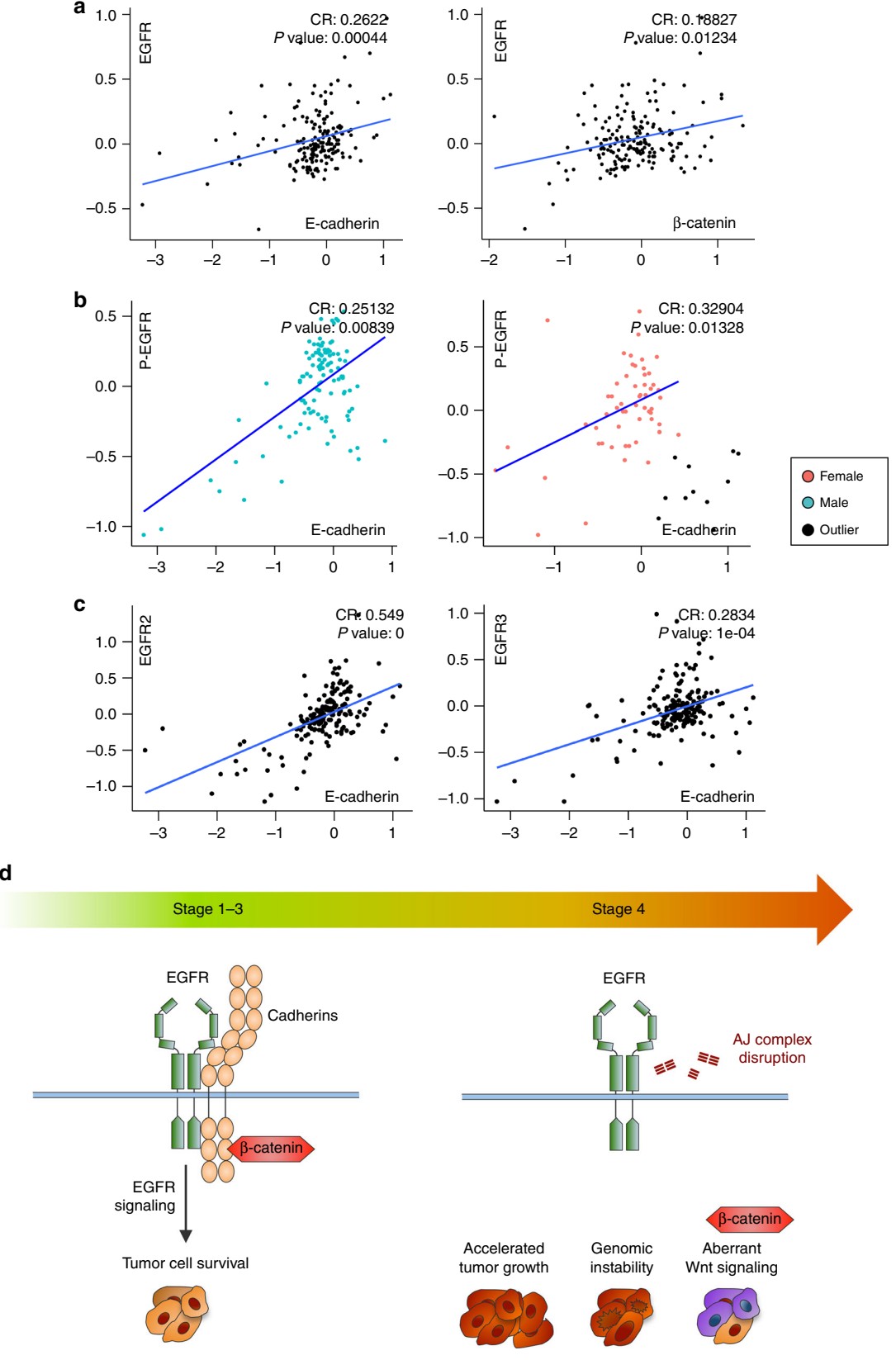

penicillin–streptomycin–glutamine. L cells (ATCC® CRL-2648) were cultured in DMEM with 10% FBS (with the addition of 0.4 mg/ml G-418, in case of L-Wnt3A (ATCC® CRL-2647)). Cell lines were tested for mycoplasma detection using an in-house PCR-based methods. For organoid culture, primary TKO HCCs were harvested and digested with an enzymatic cocktail to obtain a single-cell suspension. Organoids were generated and maintained as recently described[37]. For viral

infection, shRNA sequences were generated using psicoligomaker[61–63] and cloned into the pSicoR-PGK-RFP and psicoR-CMV-GFP vectors. Viral infection was performed as previously described[31]. To determine protein stability, cells were treated with 25 μg/ml CHX (Millipore) 24 h after plating, and harvested after the indicated treatment time period prior to immunoblotting. For Axin stabilization, TKO cells were treated with 5 μM XAV939 (Selleckchem) for 48 h[39]. For Wnt3A

**Fig. 8** Correlation between E-cadherin and epidermal growth factor receptor (EGFR) in hepatocellular carcinoma (HCC) patients. **a** Correlation between EGFR and E-cadherin (left) and β-catenin (right) protein expression in HCC patients at different stages of the disease (CR = correlation ratio; Spearman's ρ statistic). **b** Correlation between P-EGFR (Tyr1068, activated EGFR) and E-cadherin protein expression in male (left, blue dots) and female (right, red dots) HCC patients (CR = correlation ratio; Spearman's ρ statistic). Outliers are marked by black dots. See also Supplementary Fig. 8. **c** Correlation between EGFR2 (left) and EGFR3 (right) and E-cadherin protein expression in HCC patients at different stages of the disease (CR = correlation ratio; Spearman's ρ statistic). **d** In early HCC, β-catenin is recruited to the adherens junction (AJ) complex where it interacts with multiple cadherin family members. This recruitment prevents β-catenin participation in the Wnt pathway and supports HCC cell survival through the EGFR signaling pathway. Disruption of the AJ complex during HCC progression leads to increased proliferation and genomic instability, as well as activation of the Wnt signaling in patients harboring activation mutations for β-catenin

media treatment, the recipient cells were treated with media from L-Wnt3A cells mixed with fresh media at 1:1 ratio. For EGF treatment, serum-starved cells were stimulated with 20 ng/ml EGF for 15 min–1 h. For EGFR inhibition/multi-kinase inhibition, the cells were plated in a 6-well plate at $5 \times 10^5$ cells/well in the presence of DMSO or 0.1, 1, 10, or 50 μM and Afatinib, Erlotinib, or Sorafenib for 24 h and were harvested for analysis. For orthovanadate treatment, the cells were treated with 0.5 mM orthovanadate for 2 h and were harvested for analysis. To generate the growth curves, the cells were plated in triplicate in a 6-well plate at $5 \times 10^4$ cells/well and counted every 24 h for 5 days. Cells collected on days 0 and 5 were processed for RNA analysis and immunoblotting. For AnnexinV staining, the cells were plated in triplicate in a 6-well plate at $1 \times 10^5$ cells/well and cultured for 3 days prior to AnnexinV (Roche) analysis. For BrdU labeling, BrdU (1:100) were added to the media 1 h prior to harvest.

**Plasmids**. EdTC, for dnTCF4 overexpression, was a gift from Roel Nusse (Addgene plasmid #24310)[42]. Human Flag-β-catenin pcDNA3, for human β-catenin overexpression, was a gift from Eric Fearon (Addgene plasmid #16828)[64]. pWZL Blast DNE, for DNE overexpression, was obtained from Addgene (plasmid #18800)[44]; DNE sequence was further subcloned into MSCV-IRES-GFP (MigR1) vector. MigR1-GFP vector was a gift from Warren Pear (Addgene plasmid #27490)[65]. pBABE Flag-Ubiquitin, for Flag-Ubiquitin overexpression, was a gift from Donita Brady.

**Quantitative RT-PCR and sequencing analysis**. Total RNA was extracted, purified, and reverse transcribed as previously described. qPCR was performed with SYBR Green PCR Master Mix (Life Technologies) on the Viia7 Real-Time qPCR system (Life Technologies). Data were normalized using gapdh as a reference gene. For sequencing of β-catenin, total RNAs were isolated from frozen samples and reverse transcribed to obtain complementary DNAs used as a template to amplify total β-catenin[15]. Primer sequences are available in Supplementary Tables 2–3.

**Protein analysis**. For immunoprecipitation, the cells were harvested in Triton-X IP Lysis Buffer and incubated with relevant antibodies for 1 h at 4 °C, followed by 1 h incubation with A/G agarose beads. For immunoblotting, frozen tumors, control livers, and cells in culture were lysed in 1% sodium dodecyl sulfate (SDS) or radio-immunoprecipitation assay. Selected uncropped immunoblots can be found in the Supplementary Fig. 9. For IHC, standard deparaffinization, rehydration, and heat-induced epitope retrieval were performed. For MS, each sample was run on an SDS–polyacrylamide gel electrophoresis, cut into eight even slices, and further cut into 1 mm cubes by reducing, alkylating, and digesting with trypsin. Tryptic digests were analyzed by liquid chromatography with tandem MS on a hybrid LTQ Orbitrap Elite mass spectrometer. All MS/MS samples were analyzed using Sequest (Thermo Fisher Scientific), against a Uniprot Mus musculus complete proteome database (20130819, 51,891 entries) appended with common contaminants. Scaffold (version Scaffold_4.3.4, Proteome Software Inc.) was used to validate MS/MS-based peptide and protein identifications.

**Antibodies**. Mouse immunoglobulin G (IgG) (sc#2025; 3–5 μg), rabbit IgG (sc#2027; 3–5 μg), α-catenin (Sigma#c2081; 1:3000), APC (sc#896; 1:500), Axin1 (Millipore#05-1579; 1:1000), Axin1 (Cell Signaling#3323; 1:1000), β-actin (Cell Signaling#4970; 1:10,000), β-catenin (BD#610153; 1:2000-4000), phospho-β-catenin (Cell Signaling#9561; 1:1000), Claudin7 (Abcam#27487; 1:1000), E-cadherin (Cell Signaling#3195; 1:1000–3000), phospho-EGFR(Tyr1068) (Cell Signaling#2234; 1:000), EGFR (Cell Signaling#4267; 1:1000), EGFR1 (sc#373746; 1:1000), EGFR2/HER2 (sc#33684; 1:1000), Flag (Sigma#F1804; 1:5000), GS (BD#610517; 1:2000), GSK3α/β (Cell Signaling#5676; 1:000), horse radish peroxidase-conjugated Flag (Sigma#A8592; 1:5000), K-cadherin (sc#1503; 1:1000), Ki67 (BD#550609; 1:100 for IF), N-cadherin (BD#610921; 1:1000), Vimentin (Cell Signaling#5741; 1:1000), and Zeb1 (sc#H-102; 1:1000). Otherwise noted, antibody concentration listed is for immunoblotting.

**Human HCC tissue array**. Human HCC tissue array (HLiv-HCC060CD-01) was purchased from US Biomax. HCC stages were classified according to the AJCC

Clinical Staging System (v7). β-Catenin staining pattern and signal intensity were assessed by E.E. Furth, a board-certified GI tract pathologist.

**Statistical and bioinformatic analysis**. Genomic DNA sequences as well as the protein and mRNA expression (Figs. 1a, 4g, h) were obtained from TCGA browser (https://genome-cancer.ucsc.edu/proj/site/hgHeatmap/?datasetSearch = TCGA-COAD). For protein and mRNA expression, processed files (level 3) were used for further analyses and HCC patient data were separated into two groups based on protein expression. Non-parametric t test was performed to access significance between the protein and mRNA expressions of corresponding samples (Fig. 1b). Analysis of variance was performed, followed by Tukey's multiple comparisons test. The mean of each group was compared to the mean of every other group, and the p values were adjusted by the number of comparisons. (Figs. 2c, 4g, h and Supplementary Fig. 5E) The box plots encompass all data points between the 25th (lower limit) and the 75th (upper limit) percentile. The 50th percentile is indicated by an horizontal bar within the box. Lower and upper vertical bars outside the box indicate the smallest value within 1.5 times below the 25th percentile and the largest value within 1.5 times above the 75th percentile, respectively (Fig. 4f). Pearson's product-moment correlation was performed to assess correlation coefficient and significance (Fig. 5j). Data were plotted using the R package ggplot2 (ref. [66]). We used a two-sample Kolmogorov–Smirnov test to determine whether the distribution between both samples is different (Fig. 8a–c and Supplementary Fig. 8A) Protein and mRNA data was obtained from [MORPHEUS] (https://software.broadinstitute.org/morpheus/), and plotted using the R packages ggplot. Spearman's ρ statistic was used to test for association (Supplementary Fig. 8B). The histogram figure was obtained from cBioPortal Version 1.10.3-SNAPSHOT, selecting for following TCGA cancers: glioma, head and neck cancer, NSCLC, colorectal cancer, hepatobiliary cancer, ovarian cancer, and breast cancer.

**Reporting summary**. Further information on research design is available in the Nature Research Reporting Summary linked to this article.

## Data availability

The mass spectrometry data are available via Proteome Exchange with identifier PXD013095.

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

## Acknowledgements

We thank the members of the Cancer Pathobiology Division of the Department of Pathology at CHOP, as well as Warren Pear, Peter Klein, Ben Stanger, David Feldser, Nancy Speck, Wei Tong, Tom Curran, Anil Rustgi, and Christopher Lengner for their suggestions and critical reading of the manuscript. We also thank Kathryn Hamilton, Douglas Wallace, and Prasanth Potluri for their generous help with microscopy imaging. Lastly, we thank the members of the Flow Cytometry, Mass Spectrometry, and Animal Facility at CHOP. P.V. is supported by the W.W. Smith Charitable Trust Fund, a Foerderer Award, the ALSF and the Canuso foundations, the American Cancer Society (RSG-16-233-01-TBE), and start-up funds from the Center for Childhood Cancer Research at CHOP.

## Author contributions

E.K. designed, performed and analyzed experiments; A.L., C.M., N.L., and U.E. performed experiments; K.E.H. performed all bioinformatic analysis; E.E.F. analyzed all pathology-related experiments; P.V. designed, performed, and analyzed experiments and supervised the project; E.K. and P.V. wrote the manuscript.

## Additional information

**Competing interests:** The authors declare no competing interests.

