## [Peer Review File · Nature Communications]

Reviewers' comments:

Reviewer #1, Expertise: Wnt/ beta catenin (Remarks to the Author):

Canonical Wnt signaling focuses on the stability of the β catenin protein. In the absence of active receptor signaling, β -Catenin will be degraded through a dedicated destruction complex in which the Apc and Axin proteins play major roles. In the presence of receptor-bound Wnt, silencing occurs of this destruction complex. The resulting accumulation of beta Catenin leads to its nuclear transfer where it co-operates with TCF transcription factors to activate physiologically relevant Wnt target genes. A separate pool of β -Catenin is found in association with E-Cadherin, playing an important role in maintaining epithelial integrity. Aberrant expression of the complex is associated with epithelial-mesenchymal transition, fibrotic diseases and malignancies.

Pathogenic activating mutations in β -Catenin itself, or stop codons in Apc or Axin, all lead to constitutive expression of β -Catenin in the nucleus. The activation of TCF target genes has been shown in many ways to constitute the pathway towards cancer. Indeed, in the colon epithelium, this scenario is responsible for the generation of tumors.

A similar oncogenic role of β -Catenin is widely assumed for HCC (Hepato Cellular Carcinoma) too, due to the presence of a similar spectrum of Wnt-deregulating mutations in about \pm 35% of patients.

In the present manuscript this model is challenged by assuming that oncogenic β -Catenin signaling in the nucleus is restricted to late stage HCC. From the data in Figure 1 it should be clear that Wnt mutations are present from stage-I on in HCC, but that they only in stage IV lead to oncogenic actions in the nucleus. IHC for β -Catenin in tissue samples, derived from the 4 stages should support this. The absence of nuclear β -Catenin, but the presence of it at the cellular membrane in the early stages, leads to the speculation of a growth factor signaling (EGFR) role of β -Catenin in the membrane.

My main point of criticism: It has been speculated earlier that Beta catenin would not act in the nucleus, based on tissue staining results. These conclusions turned out to be wrong. It should be realized that very little nuclear beta-catenin is required for TCF target activation, much less than can be seen by staining. The gold standard of activated Wnt/TCF signaling involves scrutiny of the expression of Wnt target genes such as Axin2. Functionally, one can (inducibly) block beta-catenin-TCF signaling by conditional deletion of TCF or by inducible expression of dominant-negative TCF. Neither of these experiments are presented

To further investigate the role of membrane-bound β -Catenin, a recently developed mouse model for human HCC is exploited. In this triple Rb-protein knock-out model β -Catenin protein exclusively accumulates at the membranes of these tumor cells. In serially transplanted tumors both nuclear and membranous β -Catenin are observed. It is concluded that TKO-HCC recapitulates the evolution of β -Catenin localization in human tumors

A question that arises is, whether this assumed evolution of the role of β -Catenin, is associated with de novo Wnt mutations arising during the serial transplantation stages? If not, what type of mechanism would explain this transition?

The HCC model in fact offers an excellent opportunity to investigate the role of Wnt mutations. In vitro Introduction of an inducible dominant-negative TCF in organoid models derived from this triple Rb KO should reveal whether Wnt target genes act to promote the growth of these cells. Figure 2 presents data on lentivirally-mediated β -Catenin knock-down. This procedure leads to impaired growth in vitro and in vivo (subcutaneous transplantation). The conclusion is that the signaling role of membrane residing β -Catenin is essential, not its oncogenic role in the nucleus. The EGFR is assumed to interact with E-Cadherin, which association would be indirectly stabilized by the presence of β -Catenin in the complex. This assumption is highly speculative and is not supported by biochemical analyses, like IP/mass spec experiments showing trimolecular complexes of E-Cadherin/ β -Catenin/ and EGFR. That seems to me like an absolute minimum requirement to confirm this model.

The study heavily relies on associative-observations in various HCC databases. It, moreover, presents a highly speculative hypothesis on the role of membrane-bound β -Catenin. Although the study does not deny that Wnt deregulatory mutations can contribute to the later phase of the

disease, their effects are not investigated. The alternative growth factor receptor signaling role of β -Catenin is insufficiently supported by direct biochemical analyses.

Reviewer #2, Expertise: EGFR, liver cancer (Remarks to the Author):

In this manuscript E. Kim et. al. investigated the role of β -catenin in HCC development. Mutations in the β -catenin pathway are amongst the most abundant found in human HCC, however it is unclear whether they represent driver or cooperating events in HCC development and progression. Another unresolved issue in cell biology is how β -catenin is divided between the membrane-bound E-cadherin complex pool and the cytoplasmic/nuclear pool responsive to Wnt signaling. By analyzing human TCGA HCC datasets combined with immunohistochemistry of β -catenin in HCC tissue arrays of different stages (I-IV) the authors demonstrate that Wnt pathway mutations are present at the same frequency throughout the different stages but that at early stages β -catenin is localized at the membrane whereas at later HCC stages in the nucleus. The authors build on these findings to go on to elucidate the mechanism controlling β -catenin localization and activity in HCC using a previously generated mouse model lacking all Rb gene family members (TKO: lacking Rb, p130, p107) in the liver which develops well-differentiated HCC resembling the human disease and HCC cells derived thereof. They establish a link between β -catenin in the AJ complex and activation of EGFR signaling and suggest that this might have a critical tumor-promoting function. The findings on β -catenin localization in relation to HCC malignant stages are novel and interesting. Some mechanistic aspects could be strengthened by additional experiments specified below:

1. β -catenin expression is inhibited strongly with sh β cat3 compared to sh β cat1, which reduces β -catenin protein levels only to 50% of control (Fig. 2G). However, cell death is increased more after silencing with sh β cat1 than with sh β cat3 (Fig. 2J). How is this explained? Additionally, the results of BrdU incorporation analysis with the two shRNAs are not consistent (Fig. 2K). What is the explanation for increased G2/M transition if there is a cell cycle arrest after β -catenin knockdown with sh β cat1?
2. It is claimed that membrane-localized β -catenin escapes its degradation in TKO HCC cells. Is it possible that the half-life of β -catenin is longer in TKO HCC cells independent of its localization? Have the authors analyzed TKO cells after serial passaging in mice (e.g. Suppl. Fig. S1)?
3. Can β -catenin/E-cadherin complex be modulated to release β -catenin out from the AJ complex (e.g. like with DN E-Cad shown in Fig. 6) and force it into the Wnt pathway and can degradation vs. nuclear localization be observed? Does this affect HCC progression in vivo? What happens if TKO cells are stimulated with wnt?
4. The authors present evidence that higher E-cadherin levels negatively correlate with Wnt signaling activity in HCC. Is increased E-cadherin expression sufficient to sequester β -catenin in the AJ complex? This should be experimentally tested.
5. How do TKO induced tumors behave if treated with EGFR inhibitors in vivo? This should be analyzed also with passaged TKO cells which loose EGFR expression but might upregulate other ErbB family members.
6. It is suggested that the presence of EGFR in the AJ complex promotes HCC survival, since Afatinib treatment reduces TKO cell survival. There are several unclear issues here: Is all the EGFR in the AJ complex and how does KD of E-cad or β -catenin lead to reduced EGFR protein stability? Is there active degradation? It is surprising that all EGFR is degraded if not in the AJ complex and one wonders whether this is specific to TKO cells.
7. Minor issue: The authors show that lipocalin2 is upregulated in TKO cells similarly to what observed in human HCC and suggest that it is regulated independently of E2f. How is this result connected to the rest of the story? This should be better explained and integrated otherwise the data should be removed as it is confounding

Rebuttal letter for Kim et al, NCOMMS-18-07583

We first thank both Reviewers for their fair and rigorous assessment of our manuscript. The points that they have raised and the experiments that they suggested to address these points are appropriate and constructive. We have addressed each of these points by performing additional experiments. New Figures are highlighted in light brown in the Figures and Supplementary Figures files. We believe that the new data generated by these experiments have addressed all points raised by the Reviewers and strengthened our manuscript. We sincerely hope that the Reviewers will share our opinion and we would be happy to answer any other questions that they may have regarding the new panels displayed in this revised version of the manuscript.

Reviewer #1, Expertise: Wnt/ beta catenin (Remarks to the Author):

Canonical Wnt signaling focuses on the stability of the β catenin protein. In the absence of active receptor signaling, β -Catenin will be degraded through a dedicated destruction complex in which the Apc and Axin proteins play major roles. In the presence of receptor-bound Wnt, silencing occurs of this destruction complex. The resulting accumulation of beta Catenin leads to its nuclear transfer where it co-operates with TCF transcription factors to activate physiologically relevant Wnt target genes. A separate pool of β -Catenin is found in association with E-Cadherin, playing an important role in maintaining epithelial integrity. Aberrant expression of the complex is associated with epithelial-mesenchymal transition, fibrotic diseases and malignancies.

Pathogenic activating mutations in β -Catenin itself, or stop codons in Apc or Axin, all lead to constitutive expression of β -Catenin in the nucleus. The activation of TCF target genes has been shown in many ways to constitute the pathway towards cancer. Indeed, in the colon epithelium, this scenario is responsible for the generation of tumors.

A similar oncogenic role of β -Catenin is widely assumed for HCC (Hepato Cellular Carcinoma) too, due to the presence of a similar spectrum of Wnt-deregulating mutations in about \pm 35% of patients.

In the present manuscript this model is challenged by assuming that oncogenic β -Catenin signaling in the nucleus is restricted to late stage HCC. From the data in Figure 1 it should be clear that Wnt mutations are present from stage-I on in HCC, but that they only in stage IV lead to oncogenic actions in the nucleus. IHC for β -Catenin in tissue samples, derived from the 4 stages should support this. The absence of nuclear β -Catenin, but the presence of it at the cellular membrane in the early stages, leads to the speculation of a growth factor signaling (EGFR) role of β -Catenin in the membrane.

1) My main point of criticism: It has been speculated earlier that Beta catenin would not act in the nucleus, based on tissue staining results. These conclusions turned out to be wrong. It should be realized that very little nuclear beta-catenin is required for TCF target activation, much less than can be seen by staining. The gold standard of activated Wnt/TCF signaling involves scrutiny of the expression of Wnt target genes such as Axin2.
2) Functionally, one can (inducibly) block beta-catenin-TCF signaling by conditional deletion of TCF or by inducible expression of dominant-negative TCF. Neither of these experiments are presented

We thank the Reviewer for mentioning this critical point and we agree that more data are needed to evaluate the potential contribution of Wnt signaling to the pro-survival role identified for β -catenin in the initial version of the manuscript. To address this point, we first investigated the expression of key Wnt target genes upon repression of β -catenin expression in TKO HCC cells. As displayed in the new Figure 3I, silencing β -catenin expression in TKO HCC cells did not repress the expression of Glul and Tbx3 (two canonical Wnt target genes in the liver) but did repress the expression of Axin2. Unaltered Glul expression upon silencing β -catenin supports our initial observation that GS, the protein product of Glul and a clinical marker of Wnt activity in HCC, is not detectable in TKO HCC (Figure 3G). Therefore, repression of Axin2 expression, but not Glul and Tbx3, as well as the absence of GS expression in TKO HCC tumors indicates that Wnt signaling is active in TKO HCC, although its activity is probably limited.

To determine whether the limited Wnt activity detected in TKO HCC is responsible for the impaired cell growth and increased cell death observed upon β -catenin silencing, we performed the experiment suggested by this Reviewer and overexpressed a dominant negative form of TCF4 (DN-TCF4) in TKO HCC cells. We first found that ectopic expression of DN-TCF4 repressed the expression of Axin2 and Glul (new Figure 3K). However, TCF4 also interacts with Hnf4a on Glul promoter (Colletti et al, Gastroenterology 2009; Stanulovic et al, Hepatology 2007; Hatzis et al, MCB 2008; Gougelet et al, Hepatology 2014) and it is therefore not clear whether repression of Glul upon expression of DN-TCF4 is Wnt dependent or independent. In addition,

expression of Tbx3 and Lgr5 is not altered by the ectopic expression of DN-TCF4, overall confirming that Wnt signaling activity is limited, but not null, in TKO HCC. To directly address the important point raised by this reviewer, we found that expression of DN-TCF4 had no effect on TKO HCC cell proliferation (new Figure 3M), confirming that the impaired proliferation of TKO HCC cells triggered by silencing of β -catenin is Wnt independent.

As also suggested by this Reviewer (see the item 4) of his/her review), we sought to confirm this critical result in another experimental system. We derived organoids from freshly isolated primary TKO HCC tumors, following a protocol established by the Huch lab (first described in Huch et al, Cell, 2015). Immunofluorescence first confirmed that β -catenin is exclusively detected at the membrane of organoid cells (new Figure 2M). Similar to TKO HCC cells, β -catenin silencing represses TKO HCC organoid growth (new Figure 2N) while the expression of DN-TCF4 leaves them unaffected (new Figure 3M). From these novel data, we conclude that Wnt signaling is minimally active but it does not support TKO HCC survival and growth.

However, our data unequivocally show that Wnt signaling activating mutations occurs as early as Stage I in human HCC and the frequency of the mutations remains stable thereafter (Figure 1A), suggesting that Wnt signaling may exert a role in the early development of HCC. This concept is further supported by the fact that Wnt signaling is active, albeit in a limited manner, in TKO HCC, as demonstrated in Figure 3. We believe that experimentally identifying a potential role for Wnt signaling in early stage HCC is beyond the scope of this manuscript. However, we have modified the Discussion section of this manuscript to introduce the concept that Wnt signaling may exert discrete oncogenic activity during the early stages of HCC, such as evading immune surveillance (as already mentioned in the literature (Spranger et al, Nature 2015)).

3) To further investigate the role of membrane-bound β -Catenin, a recently developed mouse model for human HCC is exploited. In this triple Rb-protein knock-out model β -Catenin protein exclusively accumulates at the membranes of these tumor cells. In serially transplanted tumors both nuclear and membranous β -Catenin are observed. It is concluded that TKO-HCC recapitulates the evolution of β -Catenin localization in human tumors. A question that arises is, whether this assumed evolution of the role of β -Catenin, is associated with de novo Wnt mutations arising during the serial transplantation stages? If not, what type of mechanism would explain this transition?

We thank the Reviewer for this comment and we have systematically assessed the mutational status of key Wnt pathway components in serially transplanted tumors. As displayed in new Figure S3C, additional sequencing of mutational hotspots show that β -catenin remains wild-type during TKO HCC evolution. As introduced by this Reviewer, Axin and APC are also frequently targeted in HCC and we investigated their status during TKO HCC evolution. They are large proteins and, in contrast to β -catenin, can be mutated at multiple sites through their coding region to generate abnormal stop codons (as also mentioned by this Reviewer). Rather than performing extensive sequencing of these proteins, we have performed immunoblots of extracts from several control livers, primary TKO HCC tumors and serially transplanted TKO HCC tumors to detect the presence of Axin1 and APC. While APC expression is not altered during TKO HCC evolution, Axin1 expression is systematically lost in serially transplanted tumors (new Figure S3D). Therefore, we conclude from this experiment that the nuclear presence of β -catenin in serially transplanted TKO HCC tumors is a consequence of mutation(s) targeting Axin1 during TKO HCC progression, which recapitulates a phenomenon occurring in human HCC.

4) The HCC model in fact offers an excellent opportunity to investigate the role of Wnt mutations. In vitro Introduction of an inducible dominant-negative TCF in organoid models derived from this triple Rb KO should reveal whether Wnt target genes act to promote the growth of these cells.

We thank the Reviewer for this excellent suggestion. Please see our response to item 2) for the functional consequences of expressing DN-TCF4 in organoids derived from TKO HCC tumors.

5) Figure 2 presents data on lentivirally-mediated β -Catenin knock-down. This procedure leads to impaired growth in vitro and in vivo (subcutaneous transplantation). The conclusion is that the signaling role of membrane residing β -Catenin is essential, not its oncogenic role in the nucleus. The EGFR is assumed to interact with E-Cadherin, which association would be indirectly stabilized by the presence of β -Catenin in the complex. This assumption is highly speculative and is not supported by biochemical analyses, like IP/mass spec experiments showing trimolecular complexes of E-Cadherin/ β -Catenin/ and EGFR. That seems to me like an absolute minimum requirement to confirm this model.

We thank the Reviewer for this important point and we have performed additional experiments to characterize the relationship between β -catenin, E-cadherin and EGFR. In particular, we have pulled down EGFR and performed immunoblot in the unbound and bound fractions (new Figure 7A). In this experimental context, we found that EGFR interacts with both E- and N-cadherin (which elucidate the compensatory role for N-cadherin in the absence of E-cadherin, as developed in Figure 5 of the manuscript) but not β -catenin. In the initial version of the manuscript, we pulled-down β -catenin and used Mass Spectrometry to identify β -catenin interactome in an unbiased manner and had failed to detect the presence of EGFR in β -catenin interactome. Based on these experiments, we conclude that β -catenin does not directly interact with EGFR but only does so through their common interaction with cadherins. In addition, we repeated the pulled-down of EGFR followed by the detection of E-cadherin in the context of β -catenin silencing. In these conditions, we found that β -catenin silencing represses the expression of both EGFR and E-cadherin but does not impair their interaction (new Figure 7E).

6) The study heavily relies on associative-observations in various HCC databases. It, moreover, presents a highly speculative hypothesis on the role of membrane-bound β -Catenin. Although the study does not deny that Wnt deregulatory mutations can contribute to the later phase of the disease, their effects are not investigated. The alternative growth factor receptor signaling role of β -Catenin is insufficiently supported by direct biochemical analyses.

We thank the Reviewer for this suggestion. Our manuscript describes the evolving nature of β -catenin tumor promoter activity during HCC progression. While the main focus of the manuscript is on the promotion of growth factor signaling by β -catenin, we agree with this Reviewer that providing as much information as possible on the subsequent roles of β -catenin upon its relocation to the nucleus would undoubtedly strengthen it.

Marquardt et al (Hepatology, 2014) has reported that evolution of HCC to the later stage of the disease is characterized by the onset of a transcriptional program (enriched for Wnt target genes) that support cytoskeleton reorganization and EMT activity. To determine the role of β -catenin in the advanced stage of the disease, we have investigated the expression of positive and negative EMT markers at different stages of TKO HCC progression (Control, primary tumors and serial transplanted tumors). As display in the new Figure 3N, we found that Zeb1 and Vimentin, two positive markers of EMT, displayed increased expression in serially transplanted tumors compared to primary TKO HCC. In contrast, the expression of Claudin 7, a negative marker of EMT, is decreased in serially transplanted tumors compared to primary tumors. Therefore, this new results indicates a correlation between the translocation of β -catenin in the nucleus and the onset of metastasis markers during TKO HCC progression, thereby supporting the concept that activation of Wnt signaling in advanced HCC promotes metastasis.

Finally, we agree with this Reviewer that our manuscript would benefit from more mechanistic insights in the role of β -catenin in the promotion of growth factor signaling. The second Reviewer made the same comment and we therefore refer this Reviewer to the item 6) of our answer to Reviewer#2.

Reviewer #2, Expertise: EGFR, liver cancer (Remarks to the Author):

In this manuscript E. Kim et. al. investigated the role of β -catenin in HCC development. Mutations in the β -catenin pathway are amongst the most abundant found in human HCC, however it is unclear whether they represent driver or cooperating events in HCC development and progression. Another unresolved issue in cell biology is how β -catenin is divided between the membrane-bound E-cadherin complex pool and the cytoplasmic/nuclear pool responsive to Wnt signaling. By analyzing human TCGA HCC datasets combined with immunohistochemistry of β -catenin in HCC tissue arrays of different stages (I-IV) the authors demonstrate that Wnt pathway mutations are present at the same frequency throughout the different stages but that at early stages β -catenin is localized at the membrane whereas at later HCC stages in the nucleus. The authors build on these findings to go on to elucidate the mechanism controlling β -catenin localization and activity in HCC using a previously generated mouse model lacking all Rb gene family members (TKO: lacking Rb, p130, p107) in the liver which develops well-differentiated HCC resembling the human disease and HCC cells derived thereof. They establish a link between β -catenin in the AJ complex and activation of EGFR signaling and suggest that this might have a critical tumor-promoting function.

The findings on β -catenin localization in relation to HCC malignant stages are novel and interesting. Some mechanistic aspects could be strengthened by additional experiments specified below:

1. β -catenin expression is inhibited strongly with sh β cat3 compared to sh β cat1, which reduces β -catenin protein levels only to 50% of control (Fig. 2G). However, cell death is increased more after silencing with sh β cat1 than with sh β cat3 (Fig. 2J). How is this explained? Additionally, the results of BrdU incorporation analysis with the two shRNAs are not consistent (Fig. 2K). What is the explanation for increased G2/M transition if there is a cell cycle arrest after β -catenin knockdown with sh β cat1?

We thank the Reviewer for noticing this discrepancy between the efficiency of the knock-down with construct sh β -cat1 and its consequences for cell growth and cell death. We have generated new hairpins to target β -catenin expression and have added an additional efficient hairpin (sh β cat4) to study the consequence of β -catenin silencing for TKO HCC cell proliferation. We have replaced the initial panels with the new data generated by these hairpins (new Figure 2F-K).

2. It is claimed that membrane-localized β -catenin escapes its degradation in TKO HCC cells. Is it possible that the half-life of β -catenin is longer in TKO HCC cells independent of its localization? Have the authors analyzed TKO cells after serial passaging in mice (e.g. Suppl. Fig. S1)?

We thank the Reviewer for making this point. We believe that we addressed the question of β -catenin stability in the initial version of the manuscript but it appears that we poorly presented our data and we apologize to the Reviewers for this issue. Data initially presented in Figure 3 show that β -catenin half-life is long, even in the presence of a functional Wnt destruction complex. Figure 5F shows that the double silencing of E- and N-cadherin reduces the presence of β -catenin at the membrane and Figure 5E and Figure S5B shows that it is followed by a decrease in the overall protein expression (Figure 5E) and half-life (Figure S5B) of β -catenin. Unfortunately, we have not been able to derive cell lines from serially passaged tumors. However, the analysis of β -catenin phosphorylation status reveals that β -catenin is overall degraded during the entire course of HCC progression, including serially transplanted tumors (top panel in new Figure 3N). We hope that the reviewer will agree with us that the increased stability of β -catenin, as observed in TKO HCC, is a consequence of its retention at the membrane by cadherin family members.

3. Can β -catenin/E-cadherin complex be modulated to release β -catenin out from the AJ complex (e.g. like with DN E-Cad shown in Fig. 6) and force it into the Wnt pathway and can degradation vs. nuclear localization be observed? Does this affect HCC progression in vivo? What happens if TKO cells are stimulated with wnt?

We thank the Reviewer for this suggestion. We have used two strategies to disrupt the AJ complex in TKO HCC. Both silencing of E/N-cadherin as well as orthovanadate treatment impaired the stability of the AJ complex and decreased the expression of β -catenin (Figure 5E and 7F, respectively). In addition, Figure S5C shows that silencing of both E/N-cadherin does not activate Wnt target genes, indicating that release of β -catenin from the membrane per se is not sufficient to activate Wnt signaling. Based on the status of Axin1 in serially transplanted tumors (new Figure S3D; see our answer to Reviewer #1, item 3)), we propose that both disruption of the AJ complex AND inactivation of the destruction complex are required for a comprehensive activation of Wnt signaling in HCC.

Figure 5H-K and Figure S5E-I show that silencing of E/N-cadherin (and therefore disruption of the AJ complex) does not impair HCC progression in vivo upon subcutaneous injection (Figure 5H) and instead fosters the evolution of the tumor (increased cell cycle activity, increase polyploidy and mosaic activation of Wnt signaling-Figure 5I-K).

To address the consequences of stimulating with Wnt (last sentence of the Reviewer's question), we have treated TKO HCC cells with Wnt3 ligand (collected from L-cells constitutively expressing Wnt3a) and found increased GS expression (new Figure 3F), which further suggests that Wnt signaling can be activated in TKO HCC. However, new Figure 4C shows that exposure to Wnt3a does not impair the interaction between E-cadherin and β -catenin.

4. The authors present evidence that higher E-cadherin levels negatively correlate with Wnt signaling activity in HCC. Is increased E-cadherin expression sufficient to sequester β -catenin in the AJ complex? This should be experimentally tested.

We thank the reviewer for this suggestion. To address this point, we have ectopically expressed dominant negative E-cadherin (DNE) in TKO HCC cells and human HCC cells that display either high (Hep3B) or low (HepG2, SNU449) E-cadherin expression (new Figure 6D and new Figure S6D). We have deliberately used the DNE (instead of the full length version of E-cadherin) as it is sufficient to anchor β -catenin within the AJ complex at the membrane but, as it lacks the cell-cell interaction domain, and therefore cannot activate a

non-cell autonomous mechanism that could generate non-specific results regarding the localization of β -catenin. In all cell lines tested, we found that DNE pattern of expression overlaps with β -catenin and alters its subcellular localization, with the exception of Hep3B cells that already express high level of endogenous E-cadherin. We conclude from these results that expression of E-cadherin is sufficient to sequester β -catenin and potentially alter its subcellular localization.

5. How do TKO induced tumors behave if treated with EGFR inhibitors in vivo?

We thank the Reviewer for this suggestion. In vivo assessment of EGFR inhibitors in rodent models of HCC have revealed compound and sometimes conflicting consequences for HCC initiation and progression. In particular, these data show that EGF signaling inhibition impacts hepatocytes, macrophages (Kupffer cells) and hepatic stellate cells (HSC) contribution to tumor development. The literature supporting this statement is comprehensively reviewed in Komposch et Sibilja, International Journal of Molecular Sciences, 2016. Therefore, long-term treatment with EGFR inhibitor generates a compound phenotype (including increased tumor evolution) and assessing the exact and specific consequences of long-term EGF inhibition for tumor cell survival is not trivial. To address the point raised by this Reviewer, we opted to study the consequences of short-term exposure to EGF inhibitors for TKO HCC. However, the size of individual tumors and the apoptotic activity of the tumor cells are extremely variable among TKO mice developing HCC or even within one individual (this including vehicle-treated mice- please see Figure R1 below). Therefore, we have not been able to generate data that reach a rigorous level of statistical (as well as scientific) significance and we do not feel confident to display data that may appear like cherry picking in our manuscript.

Figure R1. IHC staining of cleaved caspase3. Different tumor zones of various sizes within one vehicle-treated mouse express wide ranges of apoptotic activity. Tumor zone (A) vs (B) (and (C) vs (D)) are similar in surface area but (B) and (D) express higher apoptotic activity compared to (A) and (C), respectively.

To circumvent a caveat that is inherent to the TKO HCC model, we have taken advantage of our recent capacity to generate organoids from freshly isolated HCC tumor cells. Work from the Huch lab has demonstrated that HCC-derived organoids “maintain the gene expression and genomic landscape of primary tumors and are amenable for drug testing” (Broutier et al, Nature Medicine, 2017). In addition, work performed with organoids derived from other gastro-intestinal cancers (pancreas and colorectal, reviewed in Huch et al, Development, 2017) supports similar conclusions. Importantly, organoids present the advantage of supporting the rigorous assessment of tumor cell survival upon inhibitor treatment, without an interference from other cell types found in the primary tumors (as listed above). Therefore, we have treated TKO HCC cells and freshly derived organoids with Erlotinib and Afatinib, as well as Sorafenib (which serves as a control since it targets other pathways). In this experimental setting, we found that, similar to TKO HCC derived cells, freshly isolated TKO HCC organoids are sensitive to EGFR inhibitors (new Figure 6G-H). In particular, we found that the dose needed to observe a significant impact on organoid growth is 10-fold inferior for Afatinib (1 μ M) versus Erlotinib (10 μ M). In addition, high dose (50 μ M) treatment reveals a more striking effect for Afatinib versus Erlotinib (see below for a potential explanation of this difference).

We hope that, despite the setback originating from the intrinsic variability observed in TKO HCC primary tumors, the reviewer will be satisfied by the results generated with this alternative and relevant model.

This should be analyzed also with passaged TKO cells which loose EGFR expression but might upregulate other ErbB family members.

We thank the Reviewer for this suggestion. We have interpreted (hopefully correctly) that the expression “passaged TKO cells” refers to TKO HCC derived cells and have designed two hairpins to silence EGFR expression. As displayed in new Figure 6F, the efficiency of hairpin A is quite limited (40% repression) while hairpin B efficiently represses EGFR expression to an undetectable level (top panel). Importantly, we find a dose dependent effect of EGFR silencing efficiency for TKO HCC cell growth (bottom panel). As suggested by this Reviewer, we also determined the expression level of other EGFR family members. While EGFR3 was undetectable in primary TKO HCC and TKO HCC derived cells, we found that EGFR2 expression is not increased upon EGFR silencing, suggesting against a compensatory function among family members (new Figure 6F). Analysis of EGFR2 expression during TKO HCC evolution reveals a similar pattern of expression than EGFR (new Figure 7G), indirectly suggesting that AJ complex could also regulate its activity. Accordingly, Afatinib (which targets both EGFR and EGFR2) is 10-fold more potent than Erlotinib (which only targets EGFR). Finally, correlative analysis of expression level showed a strong correlation between E-cadherin and all EGFR family members in human HCC (Figure 8A-C), suggesting that the AJ complex could indeed promote the signaling of multiple growth factors, including EGFR2-3.

6. It is suggested that the presence of EGFR in the AJ complex promotes HCC survival, since Afatinib treatment reduces TKO cell survival. There are several unclear issues here: Is all the EGFR in the AJ complex and how does KD of E-cad or β -catenin lead to reduced EGFR protein stability? Is there active degradation? It is surprising that all EGFR is degraded if not in the AJ complex and one wonders whether this is specific to TKO cells.

Both Reviewers have similarly requested more mechanistic insights into the relationship between the AJ complex and EGFR and we have merged our answer to their comments here.

As mentioned in our answer to Reviewer#1, we first identified by immunoprecipitation assay the nature of the interaction between the AJ complex and EGFR, which reveals that EGFR directly interacts with cadherin family members but not with β -catenin (new Figure 7A). The fact that silencing of E-cadherin (Figure 7B) and β -catenin (Figure 7C) has such a marked effect on EGFR expression supports the concept that most EGFR interacts with the AJ complex. We hope that the Reviewer will agree with us that determining whether ALL EGFR interacts with the AJ complex is technically challenging to determine.

As requested by this Reviewer, we have also investigated the mechanism underlying the decreased expression of EGFR upon disruption of the AJ complex. Performing cycloheximide (CHX) assay in TKO cells, we observed a rapid decrease of EGFR expression in control cells upon exposure to EGF. In contrast, EGFR expression was undetectable upon β -catenin silencing in the presence of EGF (new Figure 7H, upper panels). Without exposure to EGF, we found that EGFR expression is unchanged up to 24hrs after treatment of control cells with CHX, while EGFR expression decreases within the same time frame upon β -catenin silencing (new Figure 7H, lower panels). From this Figure, we conclude that EGFR half-life decreases upon disruption of the AJ complex. EGFR is degraded through the Lysozyme/Ubiquitin complex and we have used Ubiquitin as a marker to determine whether EGFR degradation is increased upon disruption of the AJ complex. As shown in

new Figure 7I-J, we found that EGFR is more ubiquitinated upon β -catenin silencing. New Figure 7I shows the pull-down of ectopically expressed Flag-Ubiquitin followed by the detection of EGFR in the pull down fraction. In a reverse protocol, Figure 7J display the increased formation of ubiquitin chains on EGFR upon β -catenin silencing. Collectively, these data indicate that disruption of the AJ complex impairs EGFR stability by promoting its degradation by the Lysozyme/Ubiquitin pathway.

We have performed the mechanistic studies described in this manuscript by taking advantage of complementary resources derived from the TKO HCC model (primary and transplanted tumors, derived cell lines, organoids). To expand the relevance of these studies to the human disease, we have taken advantage of TCGA human HCC datasets, to prevent any potential bias in the analysis with human HCC cell lines. The strong correlation observed in human HCC between EGFR family and E-cadherin expression suggests that the EGFR/AJ complex interaction is not specific to TKO HCC but actually represents a general mechanism to promote growth factor signaling in HCC.

7. Minor issue: The authors show that lipocalin2 is upregulated in TKO cells similarly to what observed in human HCC and suggest that it is regulated independently of E2f. How is this result connected to the rest of the story? This should be better explained and integrated otherwise the data should be removed as it is confounding

Based on this comment and the opinion of the editor, we have removed the data related to lipocalin2 from the revised version of the manuscript.

Reviewers' comments:

Reviewer #1 (Remarks to the Author):

Secondary review Kim et al, NCOMMS-18-07583

The role of Wnt signaling in HCC is widely recognized. Up to 25% of HCC patients display mutations in β -Catenin gene itself. In addition, Apc and Axin, two other components of Wnt signaling appear mutated in another 20% of HCC patients. It is commonly assumed that all these mutations are dominant oncogenic drivers of the disease.

Early observations (Cieply, B. in *Hepatology* 49, 821–831, 2009 and Prange, W. et al, in *J. Pathol.* 201, 250–259, 2003) made in HCC patients, led to the conclusion that β -Catenin exerts a dual tumor suppressor function in this disease. Sequestration of β -Catenin from its Wnt signaling role in the initial phase of HCC, and contributing to E-Cadherin-mediated cell adhesion (in Adherent Junctions, AJ) at a later phase of the disease, thereby preventing EMT and metastasis.

In this manuscript a new pro-tumorigenic role of adherent junction associated β -Catenin is promoted.

To study early phase HCC, an experimental model for HCC was exploited. In these mice HCC is generated as a consequence of targeting expression of all three Rb tumor suppressor genes (TKO-HCC). Inactivation of Rb is a hallmark of HCC and TKO-HCC clearly resembles the human liver disease.

In summary, the manuscript states that in early HCC a mechanism operates to ensure preferential integration of β -Catenin into AJ complexes. It would therefore not be available for canonical Wnt signaling. During this phase β -Catenin would be undetectable in the nucleus, but clearly accumulated in AJ complexes. In the late phase of the disease, β -Catenin would progressively move to the nucleus to perform its canonical Wnt signaling role, visualized by activation of liver-specific Wnt target genes. The tumor promoting role in the early phase would not be Wnt-related, but explained by a β -Catenin -induced stabilization of, the likewise AJ-associated, EGFR receptor.

Evaluation of experiments performed in response to the most critical remarks from primary review.

1) I criticized that histochemical absence of nuclear-localized β -Catenin does not rule out that Wnt-target genes are expressed. This can be investigated at the level of transcriptional level or functionally by introducing a dominant-negative TCF transcription factor.

In the new Figure 3 shRNA experiments, silencing β -Catenin, give support to the idea that Wnt signaling is not operating at the early phase of HCC. The results of introducing dnTCF, in addition, confirm this assumption. It, moreover, shows that survival/proliferation are not canonical Wnt-signaling dependent. Authors added an organoid model here to further substantiate this assumption.

2) I suggested that the increasing level of AJ-associated β -Catenin could be the consequence of accumulation of mutations in Wnt-signaling components, especially in serially transplanted TKO-HCC tumors.

Immunoblots were now added derived from extracts of control liver, TKO_HCC and serially transplanted TKO-HCC clearly show that Axin-1 is progressively lost in serially transplanted tumors. Authors conclude that the nuclear localization of β -Catenin in late stage TKO-HCC is the consequence of Axin-1 loss. They claim that this recapitulates the evolution of the analogous human disease. Although this may explain why nuclear β -Catenin is only detectable in late stage

disease, it does not address the mechanism of the preferred AJ-association at the early stage. The only experimental explanation that I can find in the manuscript is an increase in expression of E-Cadherin in TKO-HCC (Fig. D). Fig 5B, unfortunately, shows retention of the membrane localization of β -catenin in the presence of effective shRNA directed at E-Cadherin.

3) I asked for biochemical experiments demonstrating the interaction of the EGFR receptor with E-Cadherin and the stabilizing role of β -Catenin. Ideally, an IP/MS experiment should reveal such a tri-molecular complex. This mechanism forms the main part of the hypothesis under investigation and therefore requires solid experimental support.

In the initial version of the manuscript a β -Catenin IP/MS experiment failed to identify EGFR. IP/MS based on EGFR, performed now, identify E-Cadherin and N-Cadherin, but not β -Catenin. It is assumed that the presence of N-Cadherin alone (see point 2) is sufficient to explain the selective membrane association of β -Catenin. A remaining question is whether TKO-HCC also results in increased levels of N-Cadherin in the membrane. At page 14 of the modified manuscript, authors state that "Phosphorylation of β -Catenin by multiple tyrosine kinases on residue TYR654 disrupts its interaction with E-Cadherin. I do not understand on what experimental work this is based. It may, nevertheless, present an opportunity to validate kinase/phosphatase –mediated modifications as components of a mechanism directing β -Catenin to Cadherins in the cellular membrane. Monitoring the phosphorylation status of β -Catenin in this model would help to decide whether this post-translational modification is involved.

Although the paper has clearly gained in quality, the most important element, the alternative growth factor signaling role by AJ-associated β -catenin remains largely speculative.

Reviewer #2 (Remarks to the Author):

The authors have adequately addressed the questions raised by implementing additional experiments also employing organoid models. The paper has significantly improved and mechanistically strengthened.

We are pleased to read that Reviewer#2 is satisfied by the revised version of our manuscript. We are also happy to read that Reviewer#1 found that our revised version has clearly gained in quality. Please find below our answers to the points raised by Reviewer#1. Changes in the manuscript have been highlighted in yellow and the new Figures (found in Figure5F and Supplementary Figures section) are highlighted in magenta.

Reviewer #1 (Remarks to the Author):

The role of Wnt signaling in HCC is widely recognized. Up to 25% of HCC patients display mutations in β -Catenin gene itself. In addition, Apc and Axin, two other components of Wnt signaling appear mutated in another 20% of HCC patients. It is commonly assumed that all these mutations are dominant oncogenic drivers of the disease.

Early observations (Cieply, B. in Hepatology 49, 821–831, 2009 and Prange, W. et al, in J. Pathol. 201, 250–259, 2003) made in HCC patients, led to the conclusion that β -Catenin exerts a dual tumor suppressor function in this disease. Sequestration of β -Catenin from its Wnt signaling role in the initial phase of HCC, and contributing to E-Cadherin-mediated cell adhesion (in Adherent Junctions, AJ) at a later phase of the disease, thereby preventing EMT and metastasis.

In this manuscript a new pro-tumorigenic role of adherent junction associated β -Catenin is promoted.

To study early phase HCC, an experimental model for HCC was exploited. In these mice HCC is generated as a consequence of targeting expression of all three Rb tumor suppressor genes (TKO-HCC). Inactivation of Rb is a hallmark of HCC and TKO-HCC clearly resembles the human liver disease.

In summary, the manuscript states that in early HCC a mechanism operates to ensure preferential integration of β -Catenin into AJ complexes. It would therefore not be available for canonical Wnt signaling. During this phase β -Catenin would be undetectable in the nucleus, but clearly accumulated in AJ complexes. In the late phase of the disease, β -Catenin would progressively move to the nucleus to perform its canonical Wnt signaling role, visualized by activation of liver-specific Wnt target genes. The tumor promoting role in the early phase would not be Wnt-related, but explained by a β -Catenin -induced stabilization of, the likewise AJ-associated, EGFR receptor.

Evaluation of experiments performed in response to the most critical remarks from primary review.

1) I criticized that histochemical absence of nuclear-localized β -Catenin does not rule out that Wnt-target genes are expressed. This can be investigated at the level of transcriptional level or functionally by introducing a dominant-negative TCF transcription factor.

In the new Figure 3 shRNA experiments, silencing β -Catenin, give support to the idea that Wnt signaling is not operating at the early phase of HCC. The results of introducing dnTCF, in addition, confirm this assumption. It, moreover, shows that survival/proliferation are not canonical Wnt-signaling dependent. Authors added an organoid model here to further substantiate this assumption.

We are happy to read that this reviewer is satisfied by the additional assays performed in the revised version of the manuscript, as well as the conclusion that they support regarding the role of Wnt signaling in early stage HCC. We would like to take advantage of this opportunity to thank him for suggesting these experiments.

2) I suggested that the increasing level of AJ-associated β -Catenin could be the consequence of accumulation of mutations in Wnt-signaling components, especially in serially transplanted TKO-HCC tumors.

Immunoblots were now added derived from extracts of control liver, TKO_HCC and serially transplanted TKO-HCC clearly show that Axin-1 is progressively lost in serially transplanted tumors. Authors conclude that the nuclear localization of β -Catenin in late stage TKO-HCC is the consequence of Axin-1 loss. They claim that this recapitulates the evolution of the analogous human disease. Although this may explain why nuclear β -Catenin is only detectable in late stage disease, it does not address the mechanism of the preferred AJ-association at the early stage.

In the point 3) of his/her initial review, this reviewer wrote: "In serially transplanted tumors both nuclear and membranous β -Catenin are observed. It is concluded that TKO-HCC recapitulates the evolution of β -Catenin localization in human tumors. A question that arises is, whether this assumed evolution of the role of β -Catenin, is associated with de novo Wnt mutations arising during the serial transplantation stages? If not, what type of mechanism would explain this transition?"

We addressed that question by assessing the status of Wnt pathway components in primary and serially transplanted tumors, as explicitly requested by this reviewer. As this reviewer correctly points out in this second review, we found evidences suggesting that the nuclear localization of β -catenin in late stage HCC is the consequence of Axin-1 loss, as also occurring in patients.

However, we are confused by the summary of his/her initial comment (highlighted in bold) as well as the following statement in his/her new comment: "it does not address the mechanism of the preferred AJ-association at the early stage". In the initial review, his/her question focused on the de novo Wnt mutations arising during the serial transplantation stages and not on the "mechanism of the preferred AJ-association at the early stage", which is fundamentally different. The β -catenin field has little understanding on the mechanisms that partition β -catenin pool between the AJ complex and Wnt signaling. Determining these mechanisms, including how they are specifically disrupted in the context of HCC, is indeed very interesting but represents a very significant amount of work. In our opinion, this new request is beyond the scope of our manuscript, and in particular during a second round of revision. We sincerely regret the confusion on this point.

The only experimental explanation that I can find in the manuscript is an increase in expression of E-Cadherin in TKO-HCC (Fig. D). Fig 5B, unfortunately, shows retention of the membrane localization of β -catenin in the presence of effective shRNA directed at E-Cadherin.

We agree with this reviewer that E-cadherin up-regulation may represent a mechanism that promotes the retention of β -catenin at the membrane. However, as we developed in the discussion of our manuscript, the overarching mechanism promoting β -catenin membrane localization is probably more complex and may involve several steps, including the trafficking of β -catenin.

*This reviewer is right to point out that the silencing of E-cadherin is not sufficient to disrupt the membrane localization of β -catenin, as shown in **Figure 5B**. This finding prompted us to determine whether β -catenin interacts with several Cadherin family members in TKO HCC. Indeed, **Figures 5C-D** show that β -catenin directly interacts with E-cadherin **AND** N-cadherin in TKO HCC. Therefore, its membrane localization is the consequence of a compound activity of E- **AND** N-cadherins, as demonstrated by the fact that β -catenin is retained at the membrane in the context of E-cadherin silencing (**Figure 5B**) but is released from the membrane in the context of E- **AND** N-cadherin silencing (**Figure 5F**).*

3) I asked for biochemical experiments demonstrating the interaction of the EGFR receptor with E-Cadherin and the stabilizing role of β -Catenin. Ideally, an IP/MS experiment should reveal such a tri-molecular complex. This mechanism forms the main part of the hypothesis under investigation and therefore requires solid experimental support.

In the initial version of the manuscript a β -Catenin IP/MS experiment failed to identify EGFR. IP/MS based on EGFR, performed now, identify E-Cadherin and N-Cadherin, but not β -Catenin. It is assumed that the presence of N-Cadherin alone (see point 2) is sufficient to explain the selective membrane association of β -Catenin.

We apologize for the confusion in the presentation of our data. As explained above, it is not the presence of N-cadherin alone, but rather the combined presence of E- **AND** N-cadherin that maintains the membrane localization of β -catenin. Following the comments from this reviewer, we have made a new **Figure 5F** that also display β -catenin localization upon N-cadherin silencing. This new panel shows that the double silencing of E- and N-cadherin is required to disrupt the membrane localization of β -catenin. We thank the reviewer for making that comment.

A remaining question is whether TKO-HCC also results in increased levels of N-Cadherin in the membrane.

We thank this reviewer for this suggestion. We had previously performed RT-qPCR and immunoblot analysis to determine N-cadherin expression in TKO HCC. We did not see any significant observation that could provide additional insight into the proposed mechanism and had therefore decided not to display these data in the manuscript. However, following the suggestion of this reviewer, these data have now been included in the revised version of the manuscript (**Figure S5E-F**).

At page 14 of the modified manuscript, authors state “Phosphorylation of β -Catenin by multiple tyrosine kinases on residue TYR654 disrupts its interaction with E-Cadherin. I do not understand on what experimental work this is based.

We agree with the reviewer that this sentence is poorly constructed. The purpose of the sentence was to serve as a transition in the text and in particular as an introduction on our knowledge of the Tyrosine phosphorylation of β -catenin. Therefore we should have included a reference to the appropriate literature. Based on this reviewer’s comment, we have modified the sentence in the revised version of the manuscript.

It may, nevertheless, present an opportunity to validate kinase/phosphatase –mediated modifications as components of a mechanism directing β -Catenin to Cadherins in the cellular membrane. Monitoring the phosphorylation status of β -Catenin in this model would help to decide whether this post-translational modification is involved.

We thank the reviewer for this suggestion. **Figure 7B-C** show that genetic disruption of the β -catenin/cadherin interaction (by targeted shRNA expression) decreases the expression of EGFR. To support this result, we sought to use an alternative strategy to disrupt the β -catenin/cadherin interaction. For this purpose, **Figure 7F** shows the consequence of orthovanadate salt (a general tyrosine phosphatase inhibitor that disrupts β -catenin/cadherins interaction-Roura et al, JBC 1999) treatment for the EGFR-AJ complex. We believe that this panel is important as it confirms the results of **Figure 7B-C**, by showing that a pharmacological treatment that disrupts the β -catenin/Cadherins interaction also lead to decreased EGFR expression. Therefore, **Figure 7B-C** and **Figure 7F** use different strategies to provide mechanistic insights on the EGFR/AJ complex interaction in HCC. However, they do not prove that phosphorylation of Tyr654- β -catenin is the actual mechanism that disrupts the AJ complex during TKO HCC evolution.

The affinity of the interaction between β -catenin and E-/N-cadherins is regulated by several post-translational modifications. As already mentioned in our manuscript, phosphorylation of Tyr654 on β -catenin decreases the affinity of the interaction. In addition, phosphorylation of E-cadherin on Ser 846 and N-cadherin on Tyr 860 also decreases the affinity interaction. In contrast, phosphorylation of E-cadherin on Ser 834, 836 and 842 enhances the affinity of the interaction. Importantly, we only have a limited knowledge of the kinases and phosphatases responsible for these post-translational modifications.

In regard to the question of the reviewer regarding the phosphorylation status of β -catenin in TKO HCC, we had performed an immunoblot to detect the phosphorylation status of Tyr654- β -catenin during TKO HCC evolution. While Tyr654- β -catenin is phosphorylated in 5 out of 6 CT samples, it is only phosphorylated in 2 out of 9 TKO HCC samples, suggesting that decreased Tyr654- β -catenin phosphorylation could represent a mechanism that promotes the formation of the AJ complex in TKO HCC. However, no serial transplanted tumor samples (characterized by progressive activation of Wnt signaling) displayed a detectable phosphorylation of

*β -catenin on Tyr654, suggesting that distinct post-translational modifications may regulate the formation and the disruption of the AJ complex during TKO HCC development. The outcome of this experiment is therefore relatively open-ended as it only suggests the high complexity of the mechanism regulating the β -catenin/Cadherin interaction during HCC evolution. For this reason, we decided not to display it in our manuscript. However, based on the comment of the reviewer, we have included it as a supplementary figure (**Figure S7A**) in the revised version of the manuscript. We have also added a sentence in the discussion section.*

REVIEWERS' COMMENTS:

Reviewer #1 (Remarks to the Author):

In summary, the manuscript convincingly substantiates that in early HCC a mechanism operates to ensure preferential integration of β -Catenin into AJ complexes. The tumor promoting role in the early phase seems thus unrelated to classical Wnt signaling activity. Evidence is presented that a mechanism exists that the AJ-complexed β -Catenin helps to stabilize the presence of the EGFR receptor in the same complex. In the late phase of the disease, β -Catenin would progressively move to the nucleus to perform its canonical Wnt signaling role. The mechanism operating in this preferred traveling of β -Catenin towards the AJ-complex rather than to the nucleus is still largely unknown. The findings presented here will encourage future experiments to elucidate this important phenomenon.

Comments added

Evaluation of experiments performed in response to the most critical remarks from primary review.

1) I criticized that histochemical absence of nuclear-localized β -Catenin does not rule out that Wnt-target genes are expressed. This can be investigated at the level of transcriptional level or functionally by introducing a dominant-negative TCF transcription factor.

In the new Figure 3 shRNA experiments, silencing β -Catenin, give support to the idea that Wnt signaling is not operating at the early phase of HCC. The results of introducing dnTCF, in addition, confirm this assumption. It, moreover, shows that survival/proliferation are not canonical Wnt-signaling dependent. Authors added a organoid model here to further substantiate this assumption.

We are happy to read that this reviewer is satisfied by the additional assays performed in the revised version of the manuscript, as well as the conclusion that they support regarding the role of Wnt signaling in early stage HCC. We would like to take advantage of this opportunity to thank him for suggesting these experiments.

Good to have agreement on this !

2) I suggested that the increasing level of AJ-associated β -Catenin could be the consequence of accumulation of mutations in Wnt-signaling components, especially in serially transplanted TKO-HCC tumors.

Immunoblots were now added derived from extracts of control liver, TKO_HCC and serially transplanted TKO-HCC clearly show that Axin-1 is progressively lost in serially transplanted tumors. Authors conclude that the nuclear localization of β -Catenin in late stage TKO-HCC is the consequence of Axin-1 loss. They claim that this recapitulates the evolution of the analogous human disease. Although this may explain why nuclear β -Catenin is only detectable in late stage disease, it does not address the mechanism of the preferred AJ-association at the early stage. In the point 3) of his/her initial review, this reviewer wrote: "In serially transplanted tumors both nuclear and membranous β -Catenin are observed. It is concluded that TKO-HCC recapitulates the evolution of β -Catenin localization in human tumors. A question that arises is, whether this assumed evolution of the role of β -Catenin, is associated with de novo Wnt mutations arising during the serial transplantation stages? If not, what type of mechanism would explain this transition?"

We addressed that question by assessing the status of Wnt pathway components in primary and serially transplanted tumors, as explicitly requested by this reviewer. As this reviewer correctly

points out in this second review, we found evidences suggesting that the nuclear localization of β -catenin in late stage HCC is the consequence of Axin-1 loss, as also occurring in patients. However, we are confused by the summary of his/her initial comment (highlighted in bold) as well as the following statement in his/her new comment: "it does not address the mechanism of the preferred AJ- association at the early stage". In the initial review, his/her question focused on the de novo Wnt mutations arising during the serial transplantation stages and not on the "mechanism of the preferred AJ-association at the early stage", which is fundamentally different.

I do agree here with authors

The β -catenin field has little understanding on the mechanisms that partition β -catenin pool between the AJ complex and Wnt signaling. Determining these mechanisms, including how they are specifically disrupted in the context of HCC, is indeed very interesting but represents a very significant amount of work. In our opinion, this new request is beyond the scope of our manuscript, and in particular during a second round of revision. We sincerely regret the confusion on this point.

The only experimental explanation that I can find in the manuscript is an increase in expression of E-cadherin in TKO-HCC (Fig. D). Fig 5B, unfortunately, shows retention of the membrane localization of β -catenin in the presence of effective shRNA directed at E-cadherin. We agree with this reviewer that E-cadherin up-regulation may represent a mechanism that promotes the retention of β -catenin at the membrane. However, as we developed in the discussion of our manuscript, the overarching mechanism promoting β -catenin membrane localization is probably more complex and may involve several steps, including the trafficking of β -catenin.

We have agreement

This reviewer is right to point out that the silencing of E-cadherin is not sufficient to disrupt the membrane localization of β -catenin, as shown in Figure 5B. This finding prompted us to determine whether β -catenin interacts with several Cadherin family members in TKO HCC. Indeed, Figures 5C-D show that β -catenin directly interacts with E-cadherin AND N-cadherin in TKO HCC. Therefore, its membrane localization is the consequence of a compound activity of E- AND N-cadherins, as demonstrated by the fact that β -catenin is retained at the membrane in the context of E-cadherin silencing (Figure 5B) but is released from the membrane in the context of E- AND N-cadherin silencing (Figure 5F).

Explanation sufficiently addresses the point raised !

3) I asked for biochemical experiments demonstrating the interaction of the EGFR receptor with E-cadherin and the stabilizing role of β -Catenin. Ideally, an IP/MS experiment should reveal such a tri-molecular complex. This mechanism forms the main part of the hypothesis under investigation and therefore requires solid experimental support.

In the initial version of the manuscript a β -Catenin IP/MS experiment failed to identify EGFR. IP/MS based on EGFR, performed now, identify E-cadherin and N-cadherin, but not β -Catenin. It is assumed that the presence of N-cadherin alone (see point 2) is sufficient to explain the selective membrane association of β -Catenin.

We apologize for the confusion in the presentation of our data. As explained above, it is not the presence of N-cadherin alone, but rather the combined presence of E- AND N-cadherin that maintains the membrane localization of β -catenin. Following the comments from this reviewer, we have made a new Figure 5F that also display β -catenin localization upon N-cadherin silencing. This new panel shows that

the double silencing of E-- and N--cadherin is required to disrupt the membrane localization of β --catenin. We thank the reviewer for making that comment.

Addition of Fig 5F is clearly helpful.

A remaining question is whether TKO--HCC also results in increased levels of N--Cadherin in the membrane.

We thank this reviewer for this suggestion. We had previously performed RT--qPCR and immunoblot analysis to determine N--cadherin expression in TKO HCC. We did not see any significant observation that could provide additional insight into the proposed mechanism and had therefore decided not to display these data in the manuscript. However, following the suggestion of this reviewer, these data have now been included in the revised version of the manuscript (Figure S5E--F).

Thank you for the additional data provided !

At page 14 of the modified manuscript, authors state "Phosphorylation of β --Catenin by multiple tyrosine kinases on residue TYR654 disrupts its interaction with E--Cadherin. I do not understand on what experimental work this is based.

We agree with the reviewer that this sentence is poorly constructed. The purpose of the sentence was to serve as a transition in the text and in particular as an introduction on our knowledge of the Tyrosine phosphorylation of β --catenin. Therefore we should have included a reference to the appropriate literature. Based on this reviewer's comment, we have modified the sentence in the revised version of the manuscript.

Thanks for clearing this up

It may, nevertheless, present an opportunity to validate kinase/phosphatase --mediated modifications as components of a mechanism directing β --Catenin to Cadherins in the cellular membrane. Monitoring the phosphorylation status of β --Catenin in this model would help to decide whether this post--translational modification is involved.

We thank the reviewer for this suggestion. Figure 7B--C show that genetic disruption of the β --catenin/cadherin interaction (by targeted shRNA expression) decreases the expression of EGFR. To support this result, we sought to use an alternative strategy to disrupt the β --catenin/cadherin interaction. For this purpose, Figure 7F shows the consequence of orthovanadate salt (a general tyrosine phosphatase inhibitor that disrupts β --catenin/cadherins interaction--Roura et al, JBC 1999)) treatment for the EGFR--AJ complex. We believe that this panel is important as it confirms the results of Figure 7B--C, by showing that a pharmacological treatment that disrupts the β --catenin/Cadherins interaction also lead to decreased EGFR expression. Therefore, Figure 7B--C and Figure 7F use different strategies to provide mechanistic insights on the EGFR/AJ complex interaction in HCC. However, they do not prove that phosphorylation of Tyr654-- β --catenin is the actual mechanism that disrupts the AJ complex during TKO HCC evolution. The affinity of the interaction between β --catenin and E--/N--cadherins is regulated by several post-- translational modifications. As already mentioned in our manuscript, phosphorylation of Tyr654 on β --catenin decreases the affinity of the interaction. In addition, phosphorylation of E--cadherin on Ser 846 and N--cadherin on Tyr 860 also decreases the affinity interaction. In contrast, phosphorylation of E--cadherin on Ser 834, 836 and 842 enhances the affinity of the interaction. Importantly, we only have a limited knowledge of the kinases and phosphatases responsible for these post--translational modifications.

In regard to the question of the reviewer regarding the phosphorylation status of β --catenin in TKO HCC, we had performed an immunoblot to detect the phosphorylation status of Tyr654-- β --catenin during TKO HCC evolution. While Tyr654-- β --catenin is phosphorylated in 5 out of 6 CT samples, it is only phosphorylated in 2 out of 9 TKO HCC samples, suggesting that decreased Tyr654-- β --catenin phosphorylation could represent a mechanism that promotes the formation of the AJ complex in TKO HCC. However, no serial transplanted tumor samples (characterized by

progressive activation of Wnt signaling) displayed a detectable phosphorylation of

β -catenin on Tyr654, suggesting that distinct post-translational modifications may regulate the formation and

the disruption of the AJ complex during TKO HCC development. The outcome of this experiment is therefore relatively open-ended as it only suggests the high complexity of the mechanism regulating the β -catenin/Cadherin interaction during HCC evolution. For this reason, we decided not to display it in our manuscript. However, based on the comment of the reviewer, we have included it as a supplementary figure (Figure S7A) in the revised version of the manuscript. We have also added a sentence in the discussion section.

Thank you for investing energy in addressing this point. Unfortunately it did not provide a simple answer !